# Diffusion on language model encodings for protein sequence generation

## Abstract

Protein design necessitates a profound understanding of the intricate nature of the protein universe. While approaches based on discrete diffusion and autoregression are actively developing in the field of protein sequence generation, continuous diffusion remains underappreciated and underexplored. To address this gap, this research introduces DiMA, a latent diffusion model that leverages Gaussian diffusion on representations derived from protein language models, such as ESM-2 and CHEAP, to generate amino acid sequences. We quantitatively investigate the impact of various components of the latent diffusion model and protein encoders, revealing their contributions to enhanced protein generation performance. Additionally, we conduct an extensive evaluation of existing methods alongside DiMA using multiple metrics across two protein modalities, covering quality, novelty, diversity, and distribution matching of generated proteins. Our findings demonstrate that DiMA consistently produces novel, high-quality, and diverse protein sequences that accurately reflect the inherent structural and functional diversity of the protein space. Furthermore, we show that the proposed model can be easily adapted to address conditional tasks, such as protein family generation and inpainting. This work advances the field of protein design by providing a robust framework for latent diffusion on various protein representations, facilitating high-quality protein sequence generation.

## 1 Introduction

Generative modeling of proteins is gaining traction as a key area in academic research, potentially reshaping bioinformatics, synthetic biology, and protein-based therapeutics (Wu et al., 2021; Ovchinnikov & Huang, 2021). A key part of this research area is the focus on the generation of protein sequences or 3D models. Despite the increasing emphasis on conditional generation and family-specific fine-tuning (Madani et al., 2023; Sevgen et al., 2023), the foundational step of unconditional generation remains a challenging yet vital aspect. The reason is simple: proficiency in unconditional generation provides a solid groundwork for more specialized and nuanced conditional generation, followed by subsequent fine-tuning.

Recent advancements in generative modeling across various domains, including text, images, and video, have begun to significantly influence the field of protein generation, leading to the development of innovative approaches and methodologies. In particular, many autoregressive models have been introduced for the generation of amino acid sequences (Madani et al., 2023; Ferruz et al., 2022; Shin et al., 2021; Lv et al., 2024), demonstrating their effectiveness in capturing the complex dependencies inherent in protein sequences. In addition to autoregressive models, diffusion models have also been successfully applied to protein generation tasks. Notably, several studies (Alamdari et al., 2023; Wang et al., 2024) have adapted categorical diffusion (Austin et al., 2021) for amino acid sequence generation, effectively generalizing the ESM-2 encoder to a generative task. While significant progress has been achieved in both discrete and three-dimensional diffusion models (Watson et al., 2023; Wu et al., 2022; Lin & AlQuraishi, 2023; Fu et al., 2024), developing a Gaussian diffusion model based on continuous protein representations remains a challenging task. Some studies (Lee et al., 2023) utilize specific image-like representations of protein structures to adapt Gaussian diffusion for discrete proteins limiting their usability for other protein representations, while others (Zhang et al., 2023a) primarily focus on conditional tasks, leaving unconditional generation underexplored.

Existing studies (Lee et al., 2023; Zhang et al., 2023a) indicates that Gaussian diffusion, which has gained popularity in the realm of image processing, has yet to yield satisfactory results in the context of unconditional protein generation. This observation highlights the pressing need for more specialized approaches that are tailored to the unique characteristics and complexities of protein sequences. As the field continues to evolve, it is crucial to explore methodologies that can effectively address these challenges, ultimately enhancing our understanding of protein structure and function. Furthermore, Gaussian latent diffusion presents two notable advantages over discrete diffusion methods. First, the continuous nature of the latent space enables direct application of established score-based techniques like classifier and classifier-free guidance without requiring discrete approximations. This creates opportunities for more controlled and directed protein generation. Second, recently developed CHEAP (Lu et al., 2024) encoder, that produces a compact protein representation of both sequential and three-dimensional protein information, enables the generation of latent that produces both protein sequence and structure.

In this study, we explore Gaussian latent diffusion for protein generation and propose DiMA, a latent diffusion model based on protein language model (pLM) encodings. We investigate the use of ESM-2 (Lin et al., 2023a) and CHEAP (Lu et al., 2024) pLMs as encoders to obtain sequences of continuous encodings, upon which we train a denoising diffusion model. During inference, iterative refinement is performed, and the resulting encoding is decoded to amino acid sequence. We investigate several model components in detail: proteins encoding and decoding, diffusion model architecture, noise schedule, self-conditioning, and length sampling. Additionally, we conduct an evaluation of existing methods alongside DiMA using multiple metrics across two protein modalities, covering quality, novelty, diversity, and distribution matching of generated proteins. Furthermore, we showcase the conditional generation capabilities of our method through family specific generation and inpainting.

The main contributions of our work can be summarized as follows:

- We introduce DiMA, a diffusion-based generative model for protein sequence design. DiMA uses a latent Gaussian diffusion approach through the encodings of a protein language model.
- We investigate components of latent diffusion model for protein generation and reveal the impact of our architectural design choices and implemented techniques for effective training and sampling.
- We conduct an evaluation of existing methods alongside DiMA using multiple metrics across two protein modalities, covering quality, novelty, diversity, and distribution matching of generated proteins.
- We demonstrate that DiMA consistently produces novel, high-quality, and diverse protein sequences that accurately reflect the inherent structural and functional diversity of the protein space.

The code is available at https://anonymous.4open.science/r/DiMA-0603.

## 2 RELATED WORK

Diffusion generative models, introduced by Sohl-Dickstein et al. (2015), have gained attention for their remarkable results in image (Ho et al., 2020; Song et al., 2020b;a), and speech generation (Chen et al., 2020; Popov et al., 2021). Due to their impressive generative quality, some studies have extended the application of diffusion models to the text domain. Hoogeboom et al. (2021) and Austin et al. (2021) proposed multinomial diffusion for discrete data corruption. Subsequently, other works (Li et al., 2022; Lin et al., 2023b; Gulrajani & Hashimoto, 2023; Han et al., 2022; Strudel et al., 2022; Gao et al., 2022) adapted Gaussian diffusion to sequence learning by embedding discrete data into continuous space. Yuan et al. (2022) extended the text diffusion model to the sequence-to-sequence setting. Ye et al. (2023) conducted a study on the discrepancy of the text embedding space, demonstrating that the diffusion task at small noise scales is trivial. Zhang et al. (2023b) implemented latent text diffusion inside a VAE with an autoregressive decoder. Lovelace et al. (2022) utilized diffusion models to generate a fixed-length latent representation, mapped into a high-dimensional space with the reconstruction network before being fed into an autoregressive decoder to generate text.

In protein science, deep learning has emerged as a transformative tool. Pre-trained on extensive protein sequence datasets, it provides representations widely employed in various tasks (Elnaggar

et al., 2022; Lin et al., 2023a; Lu et al., 2024). Generative models for protein sequences, exemplified by recent advancements, enhance predictions of proteins with improved properties and functions (Wu et al., 2021; Ovchinnikov & Huang, 2021). Simultaneously, progress in the sequence-to-structure domain, as seen in models like AlphaFold (Jumper et al., 2021) and ESMFold (Lin et al., 2023a), enables the prediction of 3D protein conformation from amino acid sequences. Models such as ProteinMPNN (Dauparas et al., 2022) or ESM-IF1 (Hsu et al., 2022) predict an amino acid sequence given a specific 3D structure, effectively reverse engineering the process.

In the realm of protein generation, a diverse array of autoregressive models has been developed, establishing a sophisticated baseline for subsequent model classes (Madani et al., 2023; Ferruz et al., 2022; Shin et al., 2021; Lv et al., 2024; Hesslow et al., 2022; Shin et al., 2021). Beyond autoregressive approaches, both categorical and continuous diffusion methods have emerged as promising techniques for sequence generation (Alamdari et al., 2023; Wang et al., 2024; Lee et al., 2023; Zhang et al., 2023a). Additionally, three-dimensional diffusion models have been successfully utilized for the generation of protein structures (Watson et al., 2023; Wu et al., 2022; Lin & AlQuraishi, 2023; Fu et al., 2024). Notably, there are models that facilitate the simultaneous generation of both sequence and structure, providing a more integrated approach to protein design (Campbell et al., 2024; Ingraham et al., 2023). Furthermore, the field has seen the introduction of energy-based models (Frey et al., 2023) and generative adversarial networks (GANs) (Repecka et al., 2021), which offer alternative frameworks for protein generation.

## 3 CONTINUOUS DIFFUSION ON LM REPRESENTATIONS OF PROTEIN SEQUENCES

The proposed method comprises three parts. The first part is a pre-trained single-sequence encoder ($\mathcal{E}$) that learns a meaningful latent space corresponding to the original protein space. The second part is a diffusion model ($\mathcal{F}$) that generates vectors of protein latent space from a Gaussian noise. The third part is a decoder ($\mathcal{D}$) that maps generated latent into the sequence of amino acids.

**Encodings.** We utilize a pre-trained transformer-based pLM as an encoder with ESM-2 (Lin et al., 2023a) being the default choice unless otherwise specified. The encoder maps the sequence of discrete amino acids $y = [y_1, ..., y_s]$ of length $s$ to the latent vectors $x = [x_1, ..., x_s] \in R^{s \times d}$, $x = \mathcal{E}(y)$. Then, we employ dimension normalization to encourage each component of a single vector in the sequence $x$ to have zero mean and unit variance $z_0 = Normalize(x)$. This transformation allows us to adapt the discrete protein input to a standard Gaussian diffusion framework.

**Noise schedule.** We have found that the linear and cosine noise schedulers widely employed in the image domain (Song et al., 2020b; Ho et al., 2020; Nichol & Dhariwal, 2021) are sub-optimal for the protein domain. We conjecture that this happens due to the sequential and discrete nature of the protein representations.

The reconstruction loss of diffusion models trained with such schedulers is small at small noise scales, as shown in Figure 1 (left). Consequently, the reconstruction of $z_0$ from $z_t = \sqrt{\alpha_t} z_0 + \sqrt{1 - \alpha_t} \varepsilon$ becomes quite trivial for the model for a long period of time, leading to inefficient training. We adopted the noise schedule from (Hoogeboom et al., 2023) (sd):

$$\alpha_t = \frac{1}{1 + d^2 \tan^2(\frac{\pi t}{2})} \quad (1)$$

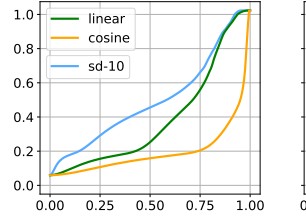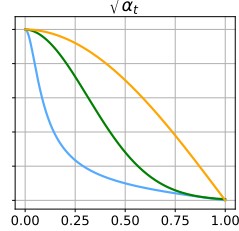

Figure 1: Left: the diffusion reconstruction loss of $z_0$ from $z_t$ with different noise schedules: $||z_0 - \hat{z}_\theta(z_t, t)||^2$. Right: $\sqrt{\alpha_t}$ equation 1.

where $d$ is a hyperparameter that reflects the rate of the schedule. The larger the value of $d$, the greater the data corruption rate. We utilize a heuristic approach based on the observation that the reconstruction loss should exhibit an approximately linear increase over diffusion time (see Figure 1). This heuristic has demonstrated improved results and aligns with the sd-10 schedule.

**Self-conditioning.** We follow recent advances in sequence generation and apply the self-conditioning technique (Chen et al., 2022) in our model. Typically, the denoising network predicts $\hat{z}_0$ using the latent variable $z_t$ and timestep $t$ as an input. Self-conditioning additionally proposes to

utilize predicted $\hat{z}_{0,s}$ from the previous timestep $s$ for estimation $\hat{z}_{0,t} = \hat{z}_\theta(z_t, t, \hat{z}_{0,s})$, $t < s$. During iterative sampling at inference, we have already computed the prediction $\hat{z}_{0,s}$ from the previous timestep. Consequently, there are no additional model launches at inference. However, we need to modify the training process so that the diffusion model trains to exploit the additional input $\hat{z}_{0,s}$.

Just like during a standard training iteration, we sample the timestep $t \sim U[0; 1]$. In half of the cases, we provide no additional input to the model, setting $\hat{z}_{0,t} = \emptyset$, where $\emptyset$ is a zero vector in our implementation. In the remaining cases, we estimate $\hat{z}_{0,t} = \hat{z}_\theta(z_t, t, \emptyset)$. Then we compute the loss:

$$\mathcal{L}(\theta) = \mathbb{E}_{\varepsilon \sim \mathcal{N}(0,\mathbf{I}), t \sim U[0;1]} \left[ ||z_0 - \hat{z}_\theta(z_t, t, \mathrm{SG}[\hat{z}_{0,t}])||^2 \right] \tag{2}$$

where $\mathrm{SG}[\cdot]$ denotes the stop-gradient operation.

This training procedure also allows sampling with zero self-condition, which is used at the first iteration of generation. Unlike the approach presented in (Chen et al., 2022) we do not concatenate $\hat{z}_{0,t}$ to $z_t$. Instead, we apply a linear transformation to $\hat{z}_{0,t}$ and incorporate it into the input of each transformer block. This modification is designed to enhance the integration of information from the denoised encodings into the transformer network, thereby improving the quality of generation. Further details regarding the architecture can be found in Appendix E.1 and Figure 14.

**Decoder.** The proposed architecture allows us to use the decoder of ESM-2 pre-trained simultaneously with the encoder on masked language modeling objectives. However, we found that additional finetuning of the decoder on a task of amino-acid reconstruction results in a more precise generation of amino acid sequences from the latents $x$ during inference. The decoder architecture comprises a single linear layer.

**Length sampling.** An important aspect of the inference phase involves determining the length of the generated sequence. There are two common approaches in this topic: padding generation and defining the sequence length prior to sampling. While many diffusion models for discrete data generate padding tokens concurrently with semantic tokens, our research indicates that using an attention mask during training is crucial for optimal performance. We conjecture that the encodings for special tokens often contain information that is meaningless from the diffusion model's perspective. Minimizing the reconstruction loss for these encodings can hinder the training process. By incorporating an attention mask, we can effectively focus the model's attention on the relevant semantic tokens, leading to more accurate and efficient generation.

During inference, the length of the generated sequence is sampled from the length distribution of the training dataset. This approach ensures that the generated sequences maintain a realistic length distribution, aligning with the characteristics of the training data (refer to Appendix E.3 for further details). Once the sequence length is determined, a random Gaussian vector is sampled. Using a fixed number of steps $T$, we iteratively generate the final $\hat{z}_0$. Following this generation process, the denormalized latent is mapped back to the corresponding amino acid sequence using the decoder. This final step completes the generation process, yielding the desired protein sequence.

**Model architecture.** We use the 12-layer Transformer model with 16 attention heads and a hidden size of 320 as a backbone for our diffusion model. We modify the model to ensure the effective operation of denoising diffusion within the specific context of protein-related data (see Appendix E.1 for more details). One noteworthy modification involves incorporating the time embedding into each transformer block. To achieve this, we use a linear projection before summation prior to each transformer block. Our experiments consistently demonstrate the effectiveness of this approach for time conditioning (refer to Section 4.2 and Table 1). An additional modification involves incorporating long skip connections (Bao et al., 2023) into the transformer model. Our practical experiments have demonstrated that this modification significantly accelerates the convergence of the model, leading to more efficient training.

## 4 EXPERIMENTS

In this section, we detail the training and validation protocols employed in our study. We then elucidate the contributions of the proposed design choices and the selection of the encoder for Gaussian latent diffusion on protein representations. Subsequently, we evaluate DiMA and existing protein sequence generative models that have open-source code, that we train under identical conditions as the proposed method. We then demonstrate that DiMA achieves the performance comparable to that of existing

pretrained models. Furthermore, we illustrate the conditional generation capabilities of our method through family-specific generation and inpainting.

We carry out experiments on two protein sequence datasets:, SwissProt (0.47M sequences) and AFDBv4-90 (Durairaj et al., 2023) (2.2M sequences), and compare our approach against a set of generative models operating directly in the amino acid sequence space. The SwissProt dataset represents a high-quality, manually curated subset of the UniProt (Consortium, 2020) database, making it an ideal choice for proof-of-concept studies due to its manageable size and high-quality annotations. Following the evaluation of our approach on the SwissProt dataset, we further assess the methods that performed best on this benchmark using the larger AFDBv4-90 dataset. Additional details regarding the dataset preprocessing steps can be found in the Appendix A.

## 4.1 EVALUATION METRICS

We conduct an evaluation of the generated sequences, employing a diverse set of metrics that collectively assess the quality, diversity, distributional matching, and novelty of the generated proteins across two modalities: sequence and structure.

To assess the **quality** of generated proteins, we employ sequence- and 3D structure-based metrics: pLDDT, pseudo-perplexity, scPerplexity, TM-score, and BLAST identity score. No single metric is sufficient for evaluating protein sequence quality; therefore, we utilize a diverse suite of complementary metrics. A key limitation of perplexity is its tendency to assign low (and thus better) values to low-information, repetitive sequences. However, such sequences typically perform poorly on our structural metrics (pLDDT, TM-score, scPerplexity), as repetitive sequences generally do not fold into stable structures. Conversely, structure-based metrics may mislead when evaluating intrinsically disordered proteins (IDPs) that lack stable 3D structures. In this context, sequence-based metrics like perplexity provide valuable complementary information. Detailed information on the computation of these quality-related metrics is available in Appendix B.1.

To further evaluate the **diversity** of the generated proteins, we employ a two-pronged approach that considers both sequence-level and cluster-level metrics. We assess the diversity of the generated amino acid sequences by quantifying the internal diversity of the amino acid sequences. Specifically, we calculate the Rep metric to penalize models with a tendency to repeatedly generate popular or commonly observed subsequences of amino acids. While such behavior may potentially inflate quality metrics, such as perplexity, it is undesirable for a robust protein generation model, as it may indicate overfitting or a lack of generative capability. In addition to sequence-level diversity, we also evaluate the cluster diversity of the generated protein samples. These metrics aim to capture the model's ability to generate a diverse set of protein clusters and a diverse range of their members, rather than concentrating the output on the most popular clusters or their prototypical representatives. The $CD_{0.5}$ metric reflects the diversity of the generated protein clusters, while the $CD_{0.95}$ metric provides insights into the diversity of the cluster representatives. By encouraging structural cluster diversity, we mitigate the risk of mode collapse. Additional details regarding the diversity metrics computation can be found in the Appendix B.2.

To examine the **distributional similarity** of the generated proteins and the test datasets, we calculate Fréchet distance (FD), maximum mean discrepancy (MMD), and 1-Wasserstein optimal transport (OT) on ProtT5 sequence representations and ProteinMPNN structure representations. These metrics are widely used for accessing in generative modeling, as they simultaneously reflect both the quality and diversity of the generated sequences. We utilize sample sizes of 2,048 sequences and compute these distributional metrics against an independent test set. Additional details on the computation of these distributional similarity metrics can be found in Appendix B.3.

To assess the **novelty** of the generated proteins, we compute the distance between each generated sequence and its nearest neighbor in the training dataset. The mean of these distances across generated sequences-Novelty aims to ensure that the generated proteins are distinct from the training proteins set. The specific details of the novelty metric computation and the distance measure employed can be found in Appendix B.4.

## 4.2 ABLATION STUDY

Existing protein generation methods based on Gaussian diffusion (Lee et al., 2023; Zhang et al., 2023a) insufficiently address the selection of optimal methodologies, largely relying on techniques

Table 1: *Ablation study of key components of DiMA-33M trained on SwissProt and AFDBv4-90 datasets using ESM-8M encoder.*

| | Model | FD-seq (↓) | FD-struct (↓) | pLDDT (↑) | Progen ppl (↓) | Rep (↓) | CD$_{0.5}$ (↑) |
|---|---|---|---|---|---|---|---|
| | Dataset | 0.13 | 0.000 | 80.7 | 6.03 | 0.045 | 1.000 |
| | Random sequences | 3.97 | 1.231 | 24.8 | 21.91 | 0.000 | 1.000 |
| SwissProt | **DiMA** | **0.38** | 0.030 | **80.8** | **5.78** | 0.250 | 0.617 |
| | w/o skip connections | 0.45 | **0.029** | 77.3 | 6.79 | 0.274 | 0.619 |
| | w/o time layers | 0.41 | 0.035 | 79.4 | 6.42 | 0.256 | 0.550 |
| | w/o ESM encoder | 1.07 | 0.069 | 62.7 | 10.42 | 0.346 | 0.619 |
| | w/o self-conditioning | 0.55 | 0.031 | 68.2 | 10.45 | 0.043 | 0.929 |
| | w/o finetuned decoder | 0.54 | 0.042 | 80.1 | 6.66 | 0.266 | 0.589 |
| | w/o length sampling | 0.67 | 0.048 | 65.0 | 11.36 | 0.050 | 0.880 |
| | w linear schedule | 0.47 | 0.031 | 77.0 | 7.66 | 0.208 | 0.611 |
| | w cosine schedule | 0.94 | 0.122 | 54.1 | 13.10 | **0.046** | 0.878 |
| | w flow-matching | 0.71 | 0.049 | 63.4 | 11.44 | 0.041 | **0.960** |
| AFDB | Dataset | 0.11 | 0.000 | 83.9 | 10.83 | 0.008 | 0.994 |
| | Random sequences | 2.55 | 1.483 | 26.2 | 22.16 | 0.000 | 1.000 |
| | **DiMA** | **0.59** | **0.033** | **73.9** | **10.44** | 0.017 | 0.994 |
| | w/o self-conditioning | 0.84 | 0.180 | 56.3 | 14.25 | 0.002 | **1.000** |

adapted from image diffusion models. In this study, we recognize the need to carefully select the diffusion components, in order to develop a Gaussian diffusion model that can effectively capture the complex patterns of protein space.

In this part of our study, we utilize the ESM-8M encoder and DiMA-33M model for our experiments. To assess the contribution of the proposed design choices to the performance of DiMA, we train several models from scratch with the following modifications:

- Removing the long **skip-connections** between the shallow and the deep transformer blocks.
- Using **time conditioning** through admixing the time embeddings to the corrupted latent vectors of amino acids instead of employing a dedicated time layer before each transformer block.
- Omitting the transformer **encoder** (ESM-2), retaining only its embedding matrix.
- Training the model without **self-conditioning**.
- Training models with **linear and cosine noise schedule**.
- Training models with padding reconstruction and without prior **length sampling**.
- Omitting **finetuning the decoder**.
- Using **flow matching** paradigm in our latent generative model.

Table 1 demonstrates that each proposed feature contributes significantly to the model's performance individually. The most substantial decrease in both the quality and distribution similarity of the generated sequences occurs in the ablated models without the ESM-2 encoder, padding omitting and length sampling, and when trained without self-conditioning. Removing skip-connections and time layers results in a less pronounced impact, but still significant decrease in repetitions of generated sequences and in a slight improvement in overall quality.

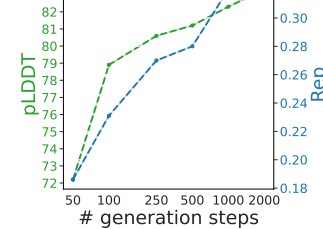

To ablate the impact of the sd-10 noise schedule, we train our diffusion model with standard linear and cosine schedules, leaving other parameters intact. We find that sd-10 significantly outperforms the cosine schedule in both quality and distribution similarity. It also achieves less expressed but better results than the linear schedule.

Figure 2: The dependence between the structural plausibility of generation, the degree of repetition, and the number of generation steps.

We demonstrate the key performance metrics in the Table 1. High correlation among distribution similarity and quality metrics led us to report only sequence and structure FID, pLDDT and Progen ppl. Complete results are provided in the Appendix C.1.

The proposed model also establishes a trade-off between structural plausibility and diversity as illustrated in Figure 2. Increasing the number of diffusion generation steps in protein production leads to higher protein quality; however, this improvement is accompanied by a slight decrease in protein diversity. This trade-off between quality and diversity provides flexibility during generation, allowing for the selection of proteins based on desired characteristics.

Table 2: *This table compares the performance of protein sequence generation using DiMA-8M and different ESM-2 encoders. Two adaptation strategies are applied: projectors addition (the first five lines) and dimensionality reduction.*

| Encoder | FD-seq ($\downarrow$) | MMD-seq ($\downarrow$) | pLDDT ($\uparrow$) | Progen ppl ($\downarrow$) | Rep ($\downarrow$) |
|---|---|---|---|---|---|
| ESM-8M | 0.541 | 0.0329 | 65.9 | 11.13 | 0.087 |
| ESM-35M | 0.338 | 0.0148 | 68.6 | 10.63 | 0.094 |
| ESM-150M | 0.270 | 0.0093 | 72.0 | 9.76 | 0.101 |
| ESM-650M | **0.266** | **0.0081** | 71.5 | 9.53 | 0.110 |
| ESM-3B | 0.279 | 0.0091 | **74.6** | **8.52** | 0.149 |
| comp ESM-150M [seq] | 2.151 | 0.2417 | 33.4 | 18.0 | 0.000 |
| comp ESM-150M [enc] | 2.387 | 0.2594 | 33.5 | 17.9 | 0.000 |

## 4.3 ENCODER STUDY

We analyze the latent spaces of all ESM-2 encoders: ESM-8M, ESM-35M, ESM-150M, ESM-650M, and ESM-3B. For this experiment, we train smaller version of DiMA, utilizing a transformer architecture with 6 layers, 16 heads, a hidden size of 320, and 8M parameters. To adapt diffusion architecture to the varying embedding dimensions of different ESM-2 encoders, we explore two approaches:

- **Dimensionality Reduction**: We attempt to compress the latent spaces of the encoders to the target dimension of 320 using two reconstruction tasks. The first task involves training a separate model to reconstruct the original encodings from the compressed representations (**comp ESM [enc]**). The second task involves training a separate model to reconstruct the original amino acid sequences from the compressed representations (**comp ESM [seq]**). After compression, the diffusion model is trained to reconstruct the compressed space of the encoder.
- **Projectors Addition**: Alternatively, we add three linear layers to our diffusion model, leaving other parameters untouched. The first linear layer projects the input $z_t$ to the dimension of 320, the second one projects self-condition $\hat{z}_{0,t}$, and last one projects the output back to the initial dimension of the encoder. In this case, the diffusion model is trained to reconstruct the initial encoder space.

These two approaches provide distinct strategies for adapting the diffusion model to varying encoder dimensions, enabling us to explore the impact of different pLM representations on the performance of protein sequence generation.

Table 2 presents a comparison of the latent spaces induced by different ESM-2 encoders, highlighting the impact of the choice of protein language model (pLM) on protein generation. The results demonstrate a clear advantage for latent spaces derived from larger encoders. While the diffusion model consistently generates higher-quality proteins when training with ESM-3B, as indicated by quality metrics, the model with ESM-650M exhibits a stronger ability to approximate the distribution of amino acid sequences in the training data. The observed behavior reflects a fundamental trade-off between quality and diversity in our current setup and reveals the interplay between model capacity and latent space complexity. DiMA with ESM-2 3B encoder shows improved quality metrics compared to 650M version (pLDDT increased from 71.5 to 74.6, ppl improved from 9.53 to 8.52), but this comes with decreased diversity ($CD_{0.5}$ drops from 0.748 to 0.660). This suggests that 8M parameter diffusion model we use for these scaling experiments is reaching a capacity limit where improvements in one aspect come at the cost of another. Given limited capacity, the model optimizes for generation quality in a smaller region of the latent space (leading to better quality metrics) rather than attempting broader but lower-fidelity coverage (yielding better distribution and diversity metrics). These findings suggest that to fully leverage larger encoders like ESM-2 3B, we likely need to scale up the diffusion model accordingly. The current results represent a capacity-constrained optimization where the model must balance quality and coverage.

The model that uses compressed representations of ESM-2 150M (Table 2, bottom) through the proposed dimensionality reduction techniques struggles to effectively learn how to generate proteins, resulting in significantly degraded performance across all evaluated metrics. These results prove this straightforward compressing technique with training additional decoder model a non-viable option for adapting a high-dimensional encoder.

These findings highlight the importance of selecting an appropriate latent space for training the diffusion model. When sufficient resources are available, we recommend using the largest available encoder to maximize the potential for generating high-quality protein sequences. This approach offers a significant advantage by allowing the utilization of the rich latent space of a large protein language model during training. However, during inference, the encoder model can be discarded, resulting in a light-weight and computationally efficient generative model. This approach effectively combines the benefits of powerful pLM representations with the power of a generative diffusion model.

## 4.4 EXPLORATION OF CHEAP ENCODER

We conduct a series of experiments using CHEAP encoder instead of ESM-2. We aim to test the possibility to apply the optimal hyperparameters discovered through our ablation studies directly to another latent space. We tested two variants: CHEAP_shorten_1_dim_64 and CHEAP_shorten_2_dim_64. Both encoders compress one dimension to 64, but CHEAP_shorten_2_dim_64 additionally reduces the sequence length dimension by half. For these experiments, we only replaced ESM-2 with CHEAP encoders while keeping all other aspects of our architecture and training procedure exactly the same. The results are remarkably strong (Tables 3, 9). Both CHEAP variants achieve impressive performance: pLDDT scores (80.3 and 81.4) closely match the dataset quality (80.7), and FD-seq metrics (0.32 and 0.36) are comparable with DiMA (0.34) while significantly outperforming other baselines. These promising results, obtained without modifications to the architecture or training procedures, support our conclusions from extensive ablation studies and demonstrate that our insights about latent diffusion for protein generation generalize well across different embedding spaces. This opens up exciting possibilities for developing new protein design models based on continuous latent diffusion. Additional results for the DiMA model using CHEAP encoders are available in Appendix C.5.

## 4.5 COMPARISON WITH BASELINE MODELS

We evaluate DiMA against various architectures for sequence generation. For a fair comparison, we train each method from scratch with the same parameter count (33M) as DiMA on the same dataset(s). During inference for models that utilize a predefined sequence length, we sample the length from the distribution of sequence lengths observed in the training set. In this experiment, we examine only methods for sequence generation with published source code.

We consider five groups of baselines. Autoregressive models: **RITA** (Hesslow et al., 2022), autoregressive transformers for the protein generation. **SeqDesign** (Shin et al., 2021), is a residual causal dilated CNN that is shown to have strong generalization capabilities over protein sequence space. **nanoGPT** (Karpathy, 2023), is a lean implementation of the GPT-2 autoregressive language model. Score-based models: **Walk-Jump** (Frey et al., 2023) method combines the contrastive divergence training of an energy-based model and improved sample quality of a score-based model. Generative adversarial networks: **ProteinGAN** (Repecka et al., 2021), a variant of the generative adversarial network in which both the discriminator and generator are CNNs based on ResNet blocks augmented with a self-attention layer. Discrete diffusion models: **EvoDiff-OADM** (Alamdari et al., 2023), a recently developed masked diffusion method. **DPLM** (Wang et al., 2024), a method that modifies ESM-2 encoders for discrete masked diffusion. **D3PM** (Austin et al., 2021), a discrete diffusion method adapted for protein generation. Flow-based models: **DFM** (Campbell et al., 2024), a recently developed discrete flow-based model for multimodel protein generation.

We estimate the characteristics of the datasets to establish reference values and define the lower expected quality associated with random sequences. We consider that optimal metric value is determined by its reference value. Consequently, a model is considered optimal when its metric value is closest to the reference value obtained from the training dataset.

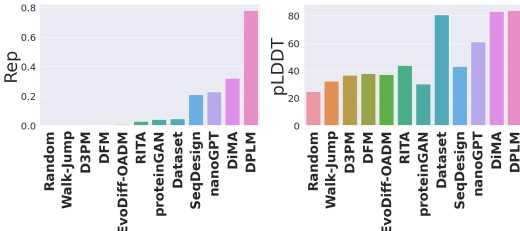

Table 3 presents the result of comparison of existing methods and DiMA. The evaluation demonstrate that DiMA produces novel, high-quality, and diverse protein sequences and displays metric values closely aligned with the reference.

Figure 3: Comparison of Rep (diversity) and pLDDT (structural quality) values for different protein generation models trained on the SwissProt dataset.

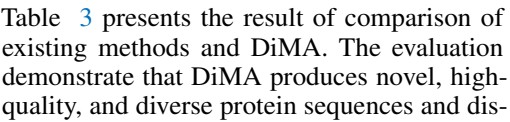

While NanoGPT, an autoregressive language

Table 3: *Performance comparison between DiMA and baseline architectures of the same parameter count trained on SwissProt and AFDB datasets. DiMA[CHEAP] refers to the implementation of DiMA using the CHEAP_shorten_1_dim_64, whereas DiMA[ESM-2] employs the ESM-2 8M encoder.*

| | Model | FD-seq ($\downarrow$) | FD-struct ($\downarrow$) | pLDDT ($\uparrow$) | Progen ppl ($\downarrow$) | Rep ($\downarrow$) | CD$_{0.5}$ ($\uparrow$) | Novelty ($\uparrow$) |
|---|---|---|---|---|---|---|---|---|
| | Dataset | 0.13 | 0.00 | 80.7 | 6.03 | 0.045 | 1.000 | 25.35 |
| | Random sequences | 3.97 | 1.23 | 24.8 | 21.91 | 0.000 | 1.000 | 85.11 |
| SwissProt | Walk-Jump | 2.63 | 0.61 | 32.4 | 15.47 | 0.001 | **1.000** | 82.20 |
| | RITA | 1.19 | 0.37 | 43.9 | 14.99 | 0.028 | 0.988 | 60.45 |
| | proteinGAN | 2.94 | 0.93 | 30.4 | 17.58 | **0.042** | 0.955 | 83.57 |
| | SeqDesign | 3.53 | 0.95 | 43.1 | 12.78 | 0.210 | 0.929 | 81.26 |
| | EvoDiff-OADM | 1.49 | 0.52 | 37.1 | 16.42 | 0.006 | 0.986 | 77.61 |
| | D3PM | 1.50 | 0.57 | 36.7 | 16.83 | 0.003 | 0.994 | 78.43 |
| | DFM | 1.46 | 0.52 | 37.8 | 16.48 | 0.004 | 0.996 | 77.27 |
| | DPLM | 0.50 | 0.15 | **84.0** | **3.57** | 0.781 | 0.494 | 11.56 |
| | nanoGPT | 1.24 | 0.15 | 61.0 | 8.87 | 0.228 | 0.900 | 53.77 |
| | DiMA [CHEAP] | **0.31** | – | 81.7 | 6.73 | 0.049 | 0.557 | 49.02 |
| | DiMA [ESM-2] | **0.34** | **0.06** | 83.3 | 5.07 | 0.320 | 0.611 | **35.74** |
| AFDB | Dataset | 0.11 | 0.00 | 83.9 | 10.83 | 0.008 | 1.000 | 57.65 |
| | Random sequences | 2.55 | 1.48 | 26.2 | 22.16 | 0.000 | 1.000 | 84.68 |
| | nanoGPT | 0.53 | 0.09 | 68.8 | 9.92 | 0.024 | **1.000** | 69.20 |
| | DPLM | 1.47 | 0.05 | **86.6** | **4.73** | 0.285 | 0.97 | 51.58 |
| | DiMA | **0.28** | **0.03** | 71.5 | 11.57 | **0.012** | **1.000** | **72.87** |

model, demonstrates promising results, it falls short of achieving the metric levels observed in the dataset. NanoGPT exhibits a lower degree of amino acid sequence repetition than DiMA, indicating greater diversity. However, it suffers from a considerable decrease in quality and proximity to the dataset's distribution, suggesting limitations in capturing the complexities of the protein space. While DPLM, a discrete diffusion model, produces proteins with high structural plausibility, it exhibits significant repetition and even duplication, indicating low diversity in generated sequences. This limitation is reflected in both distribution similarity metrics and diversity metrics. The degree of repetition is more than twice as high in DPLM compared to DiMA, while the pLDDT value shows only minor differences (see Figure 3). This suggests that DPLM, while generating structurally plausible proteins, may struggle to capture the protein space's inherent diversity effectively.

In comparison, other baselines exhibit notably poorer performance. SeqDesign and ProteinGAN, initially designed for narrow classes of proteins, may not be suitable for training on diverse datasets. While EvoDiff outperforms SeqDesign and ProteinGAN, it still demonstrates metric values closer to a random sample than to the dataset, consistent with observations in the original EvoDiff paper (Table S3 of (Alamdari et al., 2023)).

On the larger and more diverse AFDBv4-90 dataset, the performance gap between DiMA and nanoGPT narrows. DiMA achieves higher values for distributional similarity metrics, pLDDT, and Rep, while nanoGPT shows better results in Progen perplexity (9.92 against 11.57 for DiMA). Despite these achievements, both models fall short of reaching the metric values of the dataset.

DPLM exhibits a perplexity value two times lower than the dataset, suggesting a potential loss of diversity in the generated sequences. This observation is further supported by the Rep metric, which quantifies internal sequence diversity, and by the low value of the distribution similarity of sequences, indicating a limited similarity between the generated samples and the distribution of sequences in the dataset.

Additionally, we demonstrate that DiMA achieves performance comparable to that of existing pretrained large protein models. We compare the proposed model with several pretrained large protein models, including RITA (Hesslow et al., 2022), ProtGPT2 (Ferruz et al., 2022), ProGen2 (Madani et al., 2023), EvoDiff (Alamdari et al., 2023), ProLLAMA (Lv et al., 2024), DPLM (Wang et al., 2024), Chroma (Ingraham et al., 2023), Multiflow (Campbell et al., 2024), RFDiffusion (Watson et al., 2023) in different configurations. For all models, we adhere to the sampling parameters recommended by the authors. This experiment specifically focuses on methods that provide publicly accessible pretrained weights, ensuring transparency and reproducibility in our evaluation. The complete results of this experiment are provided in Appendix C.4 and Table 8.

## 4.6 CONDITIONAL GENERATION

**Family-specific generation.** Beyond the unconditional models, we also train DiMA, nanoGPT, and EvoDiff from scratch and fine-tune the SwissProt-trained models on sequences from individual protein

families. To evaluate the performance of these approaches, we use FD-seq to assess distribution similarity and pLDDT to measure the quality of the generated structures. The results are presented in Tables 11 and 10. The results demonstrate that the proposed method effectively generalizes to conditional generation, achieving high structural quality, and exhibiting strong proximity to the target distribution, suggesting that the generated sequences accurately reflect the desired properties.

**Inpainting.** We test our model in condition generation. We maske random protein sequence region (inpainted region), and model was conditioned on unmasked parts. We get SwissProt test with less than 50% sequence identity to train, as references. For each reference protein we mask region with random length (from 8 to 50 amino acids) in random position. We evaluate models by success rates. We assume that generation succeed, if generated protein has significant quality ( all sequence pLDDT $\geq$ 80 ), inpainted region is also decent (region pLDDT $\geq$ 80 ) and unmasked part does not change (predicted structure of unmasked amino acids is close to reference predicted structure, RMSD $\leq$ 1Å ). We predict pLDDT and structure by ESMFold for both generated and reference sequences. To reduce the impact of randomness in generation, we generate 10 inpaints for each reference protein. Success rate is a number of proteins where at least one attempt passes all above filters. For DiMA conditioning we add adapter (3 transformer blocks). Adapter outputs are added to all diffusion transformer blocks. We train this adapter on our unconditional train set with random region masking for 10k steps. Baselines are random and DPLM, because it can be straightforward used for this task. Both DiMA and DPLM significantly outperform random baseline (Table 12). DiMA performs slightly better than DPLM and tend to generate inpainted regions with higher pLDDT. Additionally, we evaluate the ability to generate novel regions (similar to Appendix B.4). Both baselines and DiMA produce novel regions (Inpainted region Novelty is higher than 70 %). Generation examples are located at Figure 8. These results suggest that DiMA is applicable for conditioning.

**Biological relevance.** To explore the biological relevance of the generated sequences we employ established protein annotation tool InterProScan (Paysan-Lafosse et al., 2023; Jones et al., 2014). We use three different Swissprot-trained models, DPLM, DiMA, and nanoGPT. Our analysis shows that DiMA and DPLM, models exhibiting high quality metrics, consistently generate sequences with high degree of annotation compared to the lower-performing nanoGPT (Figure 10A). This pattern is further reflected through the annotation intersections, where DiMA and DPLM demonstrate greater overlap in their annotations (Figure 10B).

While both approaches achieve similar levels of annotated proteins, they differ in their domain

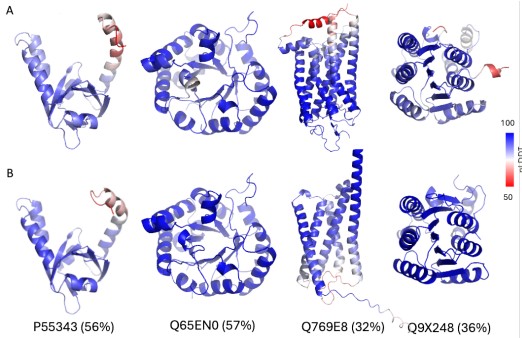

Figure 4: ESMFold predicted representative examples of proteins generated by DiMA (A) and the closest hit SwissProt (B) with UniProt IDs and the homology %, colored by pLDDT.

length characteristics. DiMA accurately reproduces dataset domain lengths and shows a tendency to generate small domains (50-75 amino acids). In contrast, DPLM frequently produces longer domains (approaching 254 amino acids in length) (Figure 10C). We hypothesize that the prevalence of long domains in DPLM correlates with its lower generation diversity, as evidenced by our diversity and distribution similarity metrics (Table 7).

## 5 CONCLUSION

In this paper, we introduce DiMA, a continuous diffusion-based model for protein sequence generation that operates within the space of protein model representations. A comprehensive ablation study quantitatively verifies the impact of DiMA's architectural features and design choices on its performance. Through extensive experiments, we evaluate the quality, diversity, distribution similarity, and biological relevance of the generated sequences. The results demonstrate that DiMA achieves comparable protein generation quality with multibillion models while utilizing a hundred times fewer parameters. Overall, this findings suggest that DiMA models are capable of generating diverse variants of natural-like proteins. The framework presented in this study provides a foundation for future research in protein generation.

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

# APPENDIX

## A  DATASETS

SwissProt is a dataset that contains a high-quality, manually annotated subset of the UniProt (Consortium, 2020) database. This dataset is small enough and good enough for proof-of-the-concept studies. After filtering out all sequences shorter than 128 and trimming all sequences longer than 254, we ended up with 470k sequences. MMseqs2 clustering of this dataset (>50% sequence identity and >80% sequence overlap) reveals the presence of clusters of similar sequences with the maximum number of sequences in a cluster equal to 1570. Each of those clusters comprises sequences that belong to a single protein family. For example, the most populous cluster is 1570 protein sequences of cytochrome b of different species, a very abundant protein involved in electron transport in eukaryotic cells. Around 120k sequences do not form clusters under the conditions used.

Another dataset we use is AFDBv4-90 from Durairaj et al. (2023), a subset of the UniRef50 database. The sequences in this dataset obey two conditions: 1. The sequence identity between all members is no more than 50%, and 2. The average predicted pLDDT by AlphaFold is no less than 90. After filtering out all sequences shorter than 128 and longer than 254, we ended up with 2.2 million whole sequences of highly diverse proteins of high quality.

## B  METRICS

### B.1  QUALITY

**pLDDT**. To assess the foldability of our generated sequences, we utilize ESMfold to predict the three-dimensional structure of the given protein sequence. For each amino acid within the predicted structure, ESMfold provides a pLDDT score, which represents the confidence of the model in the predicted positions of amino acids in the 3D structure. We average these pLDDT scores for all amino acids in the sequence to gauge the overall confidence in the predicted protein structure. It is worth noting that, while higher average pLDDT scores indicate a reliable structure prediction, lower scores may not necessarily denote poor prediction. In some cases, they can also signify the presence of intrinsically disordered regions in the protein, segments that are inherently flexible and do not

conform to a fixed structure but still play vital roles in protein functionality (Ruff & Pappu, 2021; Shukla et al., 2023).

**ProGen perplexity** To assess how probable the generated sequences we utilize the ProGen2-base (Madani et al., 2023) model of 764M parameters to estimate perplexity.

$$\mathcal{P}_{ProGen}(S) = \exp\left\{ -\frac{1}{|S|} \sum_{i=1}^{|S|} \log p(s_i|S_{<i}, \Theta_{ProGen-base}) \right\} \quad (3)$$

**ESM-2 pseudoperplexity**. To assess how probable the original sequence is under the model's distribution, we used pseudoperplexity (Salazar et al., 2019) using ESM-2 650M encoder transformer-based language model (Lin et al., 2023a). Each token (amino acid) in the sequence was masked and then predicted, considering all other tokens in the sequence. The final pseudoperplexity value is aggregated using the following equation:

$$\mathcal{P}_{ESM-2}(S) = \exp\left\{ -\frac{1}{|S|} \sum_{i=1}^{|S|} \log p(s_i|S_{\setminus i}, \Theta_{ESM-2}) \right\} \quad (4)$$

Here, $\mathcal{P}_{ESM-2}(S)$ represents the pseudoperplexity of sequence $S$, $|S|$ denotes the length of sequence $S$, $s_i$ is the $i$-th token in the sequence, $S_{\setminus i}$ represents the sequence without the $i$-th token, and $\Theta_{ESM-2}$ denotes the parameters of the ESM-2 model.

**TM-score**. To evaluate the structural relevance of the generated sequences, we turned to the TM-score (Zhang & Skolnick, 2004), a widely recognized metric for evaluating structural similarity between protein pairs. The TM-score measures the similarity between two protein structures and helps distinguish proteins with a similar fold from those with different folds. Unlike many other metrics for 3D-alignment, it does not depend on protein size and always ranges between 0 and 1, where a TM-score above 0.5 indicates a similar fold in structure. The TM-score is given by:

$$\text{TM-score} = \frac{1}{L_{\text{target}}} \sum_{i=1}^{L_{\text{query}}} \frac{1}{1 + \left(\frac{d_i}{d_0(L_{\text{target}})}\right)^2} \quad (5)$$

Here, $L_{\text{target}}$ is the length of the target protein, $L_{\text{query}}$ is the number of aligned residues between the two proteins, $d_i$ is the distance between the $i$-th aligned residue pairs, and $d_0$ is a scaling factor to normalize the length difference between query and target proteins. To calculate TM-scores for each sample of generated sequences, we first obtained their 3D structures using ESMFold. For each of these structures, we have found the closest natural protein in the SwissProt and AFDBv4-90 datasets from the AlphaFold Database (Tunyasuvunakool et al., 2021) using the FoldSeek (Van Kempen et al., 2023).

**BLAST Identity.** For each sequence, we ran BLAST with specific parameters (e-value = 0.05 and BLOSUM62 substitution matrix) to identify similar sequences within the training dataset. The number of matching amino acids between the generated sequence and the most identical sequence found in training data was normalized by sequence length and multiplied by 100 to obtain percentages. The BLAST identity metric is the average over a batch of 2048 sequences.

### B.2 DIVERSITY

**Rep**. Rep quantifies the internal diversity of generated sequences by assessing the prevalence of repeated subsequences, it is calculated as $\textbf{Rep}(y) = 1 - \prod_{n \in \{8,16,32,64\}} \frac{|\text{\# of unique n-subseq in } y|}{|\text{\# of n-subseq in } y|}$, where $y$ is a set of generated proteins. n-subseq means the subsequence of consecutive amino acids of length n.

**CD**. To evaluate the model's capacity to generate distinct protein variants while avoiding redundant outputs we employ the **clustering density** metric ($CD_t$) at two sequence identity thresholds: $t = \%50$ and $t = 95\%$. $CD_t$ represents the ratio of sequence clusters at threshold $t$ to the total number of generated proteins. Therefore, $CD_t$ ranges from 0 to 1, where 1 indicates that all sequences form

individual clusters and the sample is diverse. $CD_{0.5}$ is an established metric for assessing broad sequence diversity (Consortium, 2020), analogous to the widely-adopted TM-score threshold of 0.5 used in structure generation (Yim et al., 2023). We employ MMseqs2 (Steinegger & Söding, 2017) to perform sequence-based clustering at given thresholds $t$ (coverage = 0.8, cov-mode = 0, cluster-mode = 1.). While clustering at a moderate threshold ($50\%$) reveals the model's ability to generate diverse proteins, individual clusters may still contain nearly identical sequences—an undesirable characteristic for generative models. Therefore, we complement our analysis with $CD_{0.95}$, which specifically identifies near-duplicate sequences. This dual-threshold approach provides a more comprehensive assessment of sequence diversity compared to single-metric evaluations.

**PCD and NCD**. While $CD_t$ can capture mode collapse in a batch of sequences, it also highly rates random sequences. To evaluate the degree of novelty of the generated sequences we perform **co-clustering analysis** of generated sequences with the dataset sequences using MMseqs2 (identity = 0.5, coverage = 0.8, cov-mode = 0, cluster-mode = 1). This analysis yields two metrics: $PCD_{0.5}$ and $NCD_{0.5}$, representing the ratios of "positive" clusters (PC, containing both generated and dataset sequences) and "negative" clusters (NC, containing only generated sequences) to the total number of sequences, respectively. The desired values of $PCD_{0.5}$ and $NCD_{0.5}$ should be close to reference ones.

Notably, that generation out of distribution is also very important, so we evaluate the quality of generated sequences from other ("negative") clusters. We found that the average pLDDT of these sequences from DiMA (SwissProt and AFDBv4-90: 65 ±14 and 63 ±12, respectively), which is significantly higher than that of other models (nanoGPT: SwissProt and AFDBv4-90 43 ±12 and 52 ±16). This indicates that the model generalizes beyond the training data.

**UMAP**. To visually represent the distribution of generated sequences across PC, we trained UMAP on all sequences from PC for all models (parameters - n_neighbors - 25 and min_dict - 0.5). The UMAP plots in Figures 5 and 6 show that despite the fact that the diversity metrisc of the DIMA w/o self-conditioning are higher, DIMA with self-condition has the same coverage on the SwissProt (and even more coverage on AFDBv4-90). This and the fact that DIMA is closer to the dataset in terms of distribution learning metrics shows that DIMA w/o self condition achieved better diversity by generating sequences that greatly differ from those from the dataset.

## B.3 DISTRIBUTION SIMILARITY

**Fréchet ProtT5 Distance (FD-seq) and Fréchet ProteinMPNN Distance (FD-struct).** The Fréchet distance, also known as the 2-Wasserstein distance, quantifies the dissimilarity between two samples drawn from multivariate Gaussian distributions, denoted as $X_1 \sim \mathcal{N}(\mu_1, \Sigma_1)$ and $X_2 \sim \mathcal{N}(\mu_2, \Sigma_2)$, and is defined as follows:

$$d(X_1, X_2)^2 = ||\mu_1 - \mu_2||^2 + \text{Tr}(\Sigma_1 + \Sigma_2 - 2\sqrt{\Sigma_1 \Sigma_2}) \quad (6)$$

**Maximum mean discrepancy (MMD).** The idea behind MMD involves assessing the distance between two samples by measuring the difference in mean values resulting from applying a smooth function to the samples. A biased empirical estimate of MMD between two samples $X = \{x_1, ..., x_n\}$ and $Y = \{y_1, ..., y_n\}$ using kernel $k$ is defined as follows:

Table 4: *Review of the metrics across modalities for evaluating generation quality, diversity, novelty, and distribution similarity.*

| | **Sequence** | **Structure** |
|---|---|---|
| **Distributional similarity** | FD-seq OT-seq MMD-seq | FD-struct OT-struct MMD-struct |
| **Quality** | ProGen-2 ppl ESM-2 pppl BLAST scPerplexity | pLDDT TM-score scPerplexity |
| **Diversity** | Rep CD | |
| **Novelty** | Novelty | |

$$MMD_k^2(X, Y) = \frac{1}{n^2} \sum_{i=1}^{n} \sum_{j=1}^{n} \left( k(x_i, x_j) + k(y_i, y_j) - 2k(x_i, y_j) \right) \quad (7)$$

As a kernel function, we used the radial basis function kernel (RBF). We evaluated the distance between batches of sequences, each of size $n$ equal to 2048, sampled from the dataset and generated

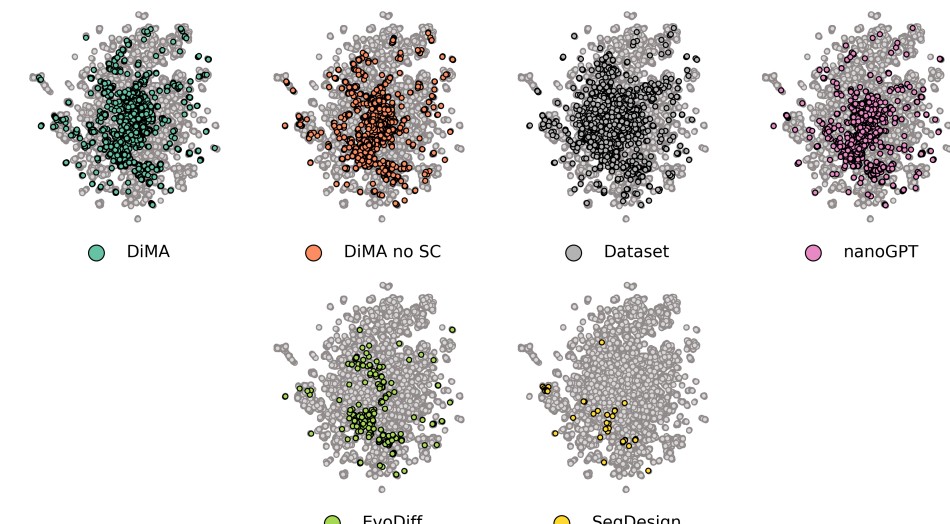

Figure 5: UMAP projection of sequences from PC. Training dataset - SwissProt. Grey background points - dataset sequences from PC.

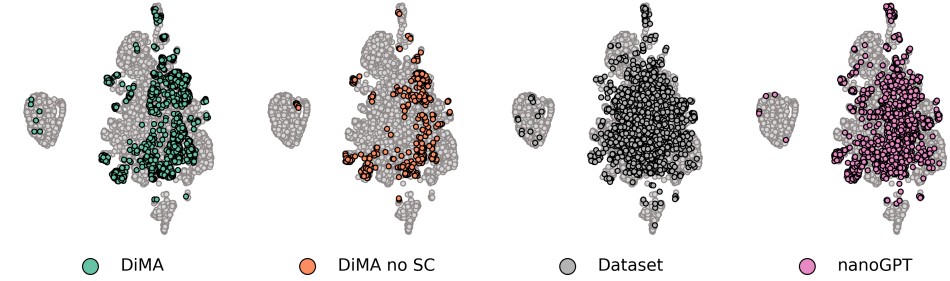

Figure 6: UMAP projection of sequences from PC. Training dataset - AFDBv4-90. Grey background points - dataset sequences from PC.

by the respective models. Following the methodology proposed for 3D structures in (Joshua Southern & Correia, 2023), we utilized ProtT5 sequence representations to calculate MMD.

**1-Wasserstein optimal transport (OT)**. The BLAST identity metric effectively evaluates the similarity between generated and natural sequences. However, its limitation lies in assessing the model's capability to produce diverse sequences, as it may identify the same dataset sequence as the closest match for every generated sequence. To overcome this limitation, we employ transportation theory to establish optimal pairs between generated sequences and the dataset.

Optimal transport theory, initially devised for solving economic problems, has found applications in various fields, including physics, biology, and tomography. To implement our approach, we calculate pairwise Levenshtein distances and use them as transportation costs. Subsequently, we determine optimal sequence pairs using the Earth Mover Distance (EMD) solver with a uniform distribution of the samples. We use the average distance between these optimal pairs, measuring both the diversity and proximity of generated samples to the dataset.

The inherent diversity of the dataset, i.e., when a sample from the dataset pairs with itself, gives zero distances ($OT(dataset) = 0$). In contrast, random sequences form optimal pairs with the highest mean distances, as illustrated in Figure 7. The optimal transport distance distributions reveal differences in how models capture the protein sequence space. Most ablation studies (Figure 7, center) show distributions similar to the reference, except for flow matching, cosine scheduler, and no encoder variants, indicating these components are most critical for DiMA's performance. Several baseline models (D3PM, DFM, EvoDiff-OADM, RITA) cluster around a similar mode between random and

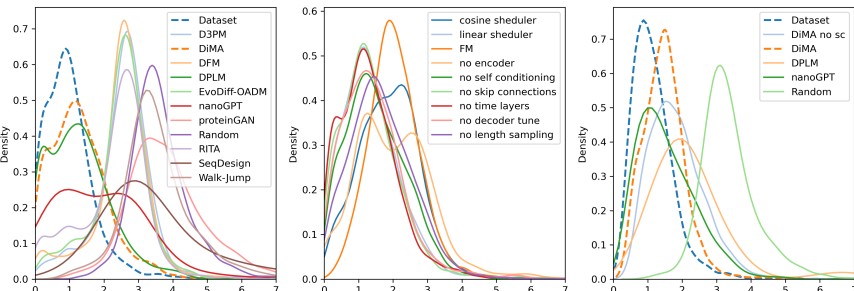

Figure 7: The distribution of optimal transport distances between pairs of generated and dataset sequences. For each model, we compute pairwise Levenshtein distances between generated sequences and dataset sequences, then find optimal matching pairs using Earth Mover Distance with uniform distribution of samples. **Left**: Comparison of DiMA against baseline models on SwissProt dataset. **Center**: Analysis of DiMA's architectural components through ablation studies. **Right**: Performance comparison on the larger AFDBv4-90 dataset. The dashed blue line represents the reference distribution obtained by matching samples within the dataset (optimal transport distance to itself), while the dashed orange line shows DiMA's distribution.

reference distributions, suggesting they mainly learn basic patterns like amino acid frequencies while capturing only a limited set of protein families, as evidenced by their left tail behavior (Figure 7, left).

DiMA's distribution (dashed orange) most closely matches the dataset reference (dashed blue) across both datasets. On SwissProt, DPLM shows a sharp, concentrated peak indicating high-quality but limited diversity, while other baselines show broader, right-shifted distributions indicating greater deviation from natural sequences. On the larger AFDBv4-90 dataset, while nanoGPT's distribution mode is closer to the reference, DiMA generates fewer distant proteins (smaller right tail) and better maintains the overall distribution shape, demonstrating robust performance even with increased dataset complexity (Figure 7 right).

Although our OT implementation offers advantages over BLAST, it has a special feature: the EMD solver identifies an exact pair for each sequence. This poses a challenge when dealing with two query sequences that are similar to one dataset sequence but distant from others, resulting in one close pair and one distant pair. However, we employ EMD precisely to penalize such cases, reinforcing the generation of diverse rather than similar sequences.

**Structural analogues**. To measure structural distribution similarity, we calculate analogous FD, MMD, OT metrics using structural encoder ProteinMPNN. ProteinMPNN is a powerful graph neural network (GNN) model pretrained on a massive dataset of protein structures.

### B.4 NOVELTY

To directly evaluate the potential memorization of the training data, we measure **novelty** by calculating the mean sequence identity between each generated sequence and its nearest neighbor in the training dataset.

We assume that novel proteins should be far from the train dataset, so for each generated sequence, we computed distance to the nearest train sequence. The golden standard for pairwise distance measure between amino acid sequences is an alignment score using Needleman–Wunsch (NW) algorithm. However, due to $O(N^2)$ calculation cost we use BLAST to find the nearest sequence in the training set and only then we align these sequences using NW. (We employ BLAST and NW with the following parameters: evalue = 15.05, matrix = BLOSUM62, word_size= 2; matrix= BLOSUM62, gap_open= -10, gap_extend=0.5). The novelty value of a batch of generated sequences is defined as **Novelty**$(y) = \frac{1}{s} \sum_{i=1}^{s} 1 - \frac{|\text{\# of same letters in alignment}|}{\text{alignment length}}$, where $y$ is a set of generated proteins $s$

Table 5: *Comprehensive set of metrics assessing the quality, distribution matching, and diversity of the generated proteins for model evaluation in the ablation study. 250 steps were used during generation.*

| | Model | pLDDT (↑) | Progen ppl (↓) | ESM-2 pppl (↓) | scPpl (↓) | TM-score (↑) | BLAST (↑) |
|---|---|---|---|---|---|---|---|
| **Quality** / SwissProt | Dataset | 80.7 | 6.03 | 5.35 | 1.88 | 0.80 | 100 |
| | Random sequences | 24.8 | 21.91 | 21.53 | 2.77 | 0.33 | 0 |
| | **DiMA** | **80.8** | **5.78** | **5.20** | **1.80** | **0.85** | **68** |
| | w/o skip connections | 77.3 | 6.79 | 5.84 | 1.87 | 0.82 | 61 |
| | w/o time layers | 79.4 | 6.42 | 5.49 | 1.83 | 0.85 | 66 |
| | w/o ESM encoder | 62.7 | 10.42 | 9.22 | 2.09 | 0.71 | 48 |
| | w/o self-conditioning | 68.2 | 10.45 | 9.18 | 2.08 | 0.74 | 46 |
| | w/o finetuned decoder | 80.1 | 6.66 | 5.59 | 1.78 | 0.85 | 65 |
| | w/o length sampling | 65.0 | 11.36 | 9.84 | 2.12 | 0.72 | 44 |
| | w linear schedule | 77.0 | 7.66 | 6.29 | 1.89 | 0.82 | 58 |
| | w cosine schedule | 54.1 | 13.11 | 10.86 | 2.16 | 0.60 | 34 |
| | w flow-matching | 63.5 | 11.44 | 8.97 | 2.08 | 0.68 | 40 |
| AFDB | Dataset | 83.9 | 10.83 | 5.79 | 1.75 | 0.92 | 100 |
| | Random sequences | 26.2 | 22.16 | 21.67 | 2.75 | 0.35 | 0 |
| | **DiMA** | **73.9** | **10.44** | 8.50 | **1.90** | **0.85** | **48** |
| | w/o self-conditioning | 56.3 | 14.25 | 12.08 | 2.18 | 0.69 | 31 |

| | | FD-seq (↓) | MMD-seq (↓) | OT-seq (↓) | FD-struct (↓) | MMD-struct (↓) | OT-struct (↓) |
|---|---|---|---|---|---|---|---|
| **Distributional Similarity** / SwissProt | Dataset | 0.13 | 0.000 | 1.08 | 0.000 | 0.000 | 0.053 |
| | Random sequences | 3.97 | 0.200 | 3.88 | 1.231 | 0.412 | 1.313 |
| | **DiMA** | **0.38** | **0.016** | **1.26** | **0.030** | 0.004 | 0.090 |
| | w/o skip connections | 0.45 | 0.021 | 1.36 | **0.029** | **0.002** | **0.081** |
| | w/o time layers | 0.41 | 0.022 | 1.29 | 0.035 | 0.004 | 0.097 |
| | w/o ESM encoder | 1.07 | 0.068 | 2.04 | 0.069 | 0.010 | 0.153 |
| | w/o self-conditioning | 0.55 | 0.047 | 1.51 | 0.031 | 0.005 | 0.093 |
| | w/o finetuned decoder | 0.54 | 0.031 | 1.44 | 0.042 | 0.004 | 0.108 |
| | w/o length sampling | 0.67 | 0.058 | 1.67 | 0.048 | 0.007 | 0.139 |
| | w linear schedule | 0.47 | 0.026 | 1.37 | 0.031 | 0.003 | 0.092 |
| | w cosine schedule | 0.94 | 0.091 | 1.90 | 0.122 | 0.019 | 0.215 |
| | w flow-matching | 0.71 | 0.063 | 1.75 | 0.049 | 0.008 | 0.130 |
| AFDB | Dataset | 0.11 | 0.001 | 1.15 | 0.000 | 0.000 | 0.052 |
| | Random sequences | 2.55 | 0.339 | 3.41 | 1.483 | 0.133 | 1.554 |
| | **DiMA** | **0.59** | **0.044** | 1.50 | **0.033** | **0.002** | **0.110** |
| | w/o self-conditioning | 0.85 | 0.089 | 1.88 | 0.180 | 0.015 | 0.263 |

| | | Rep (↓) | $CD_{0.5}$ (↑) | $CD_{0.95}$ (↑) | $PCD_{0.5}$ (↑) | $NCD_{0.5}$ |
|---|---|---|---|---|---|---|
| **Diversity** / SwissProt | Dataset | 0.045 | 1.000 | 0.943 | 0.990 | 0.304 |
| | Random sequences | 0.000 | 1.000 | 1.000 | 1.000 | 0.000 |
| | **DiMA** | 0.250 | 0.617 | 0.996 | 0.246 | 0.392 |
| | w/o skip connections | 0.274 | 0.619 | 0.990 | 0.187 | 0.439 |
| | w/o time layers | 0.256 | 0.550 | 1.000 | 0.246 | **0.347** |
| | w/o ESM encoder | **0.346** | 0.619 | 1.000 | 0.107 | 0.507 |
| | w/o self-conditioning | 0.043 | **0.929** | 1.000 | 0.146 | 0.779 |
| | w/o finetuned decoder | 0.266 | 0.589 | 0.996 | **0.255** | 0.357 |
| | w/o length sampling | 0.050 | 0.880 | 1.000 | 0.089 | 0.726 |
| | w linear schedule | 0.208 | 0.611 | 1.000 | 0.181 | 0.431 |
| | w cosine schedule | 0.046 | 0.878 | 1.000 | 0.017 | 0.798 |
| | w flow-matching | 0.041 | 0.960 | 1.000 | 0.214 | 0.945 |
| AFDB | Dataset | 0.008 | 0.994 | 1.000 | 0.029 | 0.966 |
| | Random sequences | 0.000 | 1.000 | 1.000 | 0.000 | 1.000 |
| | **DiMA** | **0.017** | 0.994 | **1.0** | **0.002** | **0.992** |
| | w/o self-conditioning | 0.002 | 1.000 | 1.000 | 0.000 | 1.000 |

# C ADDITIONAL RESULTS

## C.1 ABLATION STUDY ON SWISSPROT AND AFDBV4-90DATASETS

This section provides a comprehensive analysis of the quality, diversity, and distribution matching of the generated proteins, utilizing additional metrics to facilitate a thorough evaluation of the models in the ablation study. The results of this analysis are detailed in Table 5.

## C.2 ENCODER STUDY ON SWISSPROT DATASETS

This section provides an in-depth analysis of the quality, diversity, and distribution similarity of the generated proteins, incorporating additional metrics to deliver a thorough evaluation of the models trained with various ESM-2 encoders. The results of this analysis are detailed in Tables 6

Table 6: *The complete results for DiMA-8M evaluation utilising various ESM-2 encoders*

| | Encoder | pLDDT ($\uparrow$) | Progen ppl ($\downarrow$) | ESM-2 pppl ($\downarrow$) | scPpl ($\downarrow$) | BLAST ($\uparrow$) |
|---|---|---|---|---|---|---|
| **Quality** | ESM-8M | 65.9 | 11.13 | 7.99 | 2.09 | 44 |
| | ESM-35M | 68.6 | 10.63 | 7.30 | 2.04 | 47 |
| | ESM-150M | 72.1 | 9.76 | 6.48 | 1.98 | 51 |
| | ESM-650M | 71.5 | 9.53 | 6.18 | 1.98 | 51 |
| | ESM-3B | 74.6 | 8.52 | 5.71 | 1.91 | 56 |
| | comp ESM-150M [ce] | 33.4 | 17.95 | 17.89 | 2.55 | 3 |
| | comp ESM-150M [mse] | 33.5 | 17.91 | 16.89 | 2.54 | 3 |

| | | FD-seq ($\downarrow$) | MMD-seq ($\downarrow$) | OT-seq ($\downarrow$) |
|---|---|---|---|---|
| **Distributional Similarity** | ESM-8M | 0.541 | 0.0329 | 2.53 |
| | ESM-35M | 0.338 | 0.0148 | 2.26 |
| | ESM-150M | 0.270 | 0.0093 | 2.15 |
| | ESM-650M | 0.266 | 0.0081 | 2.21 |
| | ESM-3B | 0.279 | 0.0091 | 2.17 |
| | comp ESM-150M [seq] | 2.151 | 0.2417 | 3.53 |
| | comp ESM-150M [enc] | 2.387 | 0.2594 | 3.82 |

| | | Rep ($\downarrow$) | $CD_{0.5}$ ($\uparrow$) | $CD_{0.95}$ ($\uparrow$) | $PCD_{0.5}$ ($\uparrow$) | $NCD_{0.5}$ |
|---|---|---|---|---|---|---|
| **Diversity** | ESM-8M | 0.087 | 0.773 | 1.000 | 0.130 | 0.617 |
| | ESM-35M | 0.094 | 0.775 | 1.000 | 0.218 | 0.546 |
| | ESM-150M | 0.101 | 0.777 | 1.000 | 0.269 | 0.501 |
| | ESM-650M | 0.110 | 0.748 | 1.000 | 0.291 | 0.464 |
| | ESM-3B | 0.149 | 0.660 | 0.998 | 0.304 | 0.359 |
| | comp ESM-150M [seq] | 0.000 | 1.000 | 1.000 | 0.000 | 1.000 |
| | comp ESM-150M [enc] | 0.000 | 1.000 | 1.000 | 0.000 | 1.000 |

## C.3 COMPARISON WITH BASELINE MODELS ON SWISSPROT AND AFDBV4-90 DATASETS

This section presents an expanded analysis of the quality, diversity, and distribution similarity of the generated proteins, exploring additional metrics to provide a comprehensive evaluation of the DiMA and baselines. The results are presented in Tables 7.

## C.4 COMPARISON WITH PRE-TRAINED PROTEIN MODELS

In this section we compare DiMA with existing large protein models, including RITA (Hesslow et al., 2022), ProtGPT2 (Ferruz et al., 2022), ProGen (Madani et al., 2023), EvoDiff (Alamdari et al., 2023), ProLLAMA (Lv et al., 2024), DPLM (Wang et al., 2024), Chroma (Ingraham et al., 2023), Multiflow (Campbell et al., 2024), RFDiffusion (Watson et al., 2023) in different configurations. For all models, we adhere to the sampling parameters recommended by the authors. This experiment specifically focuses on methods that provide publicly accessible pretrained weights, ensuring transparency and reproducibility in our evaluation.

The majority of models were pre-trained on distinct versions of the UniProt (Consortium, 2020) dataset. As a result, the application of distributional similarity metrics in the current experiment is rendered unfeasible. Consequently, we focused solely on evaluating quality and diversity metrics. Given that RFDiffusion generates protein structures, we employed ProteinMPNN, a neural network trained to predict amino acid sequences from 3D protein structures, to infer sequences from the generated structures. The authors of RFDiffusion ran ProteinMPNN multiple times for each generated structure and selected the sequence with the lowest perplexity as the final prediction. In contrast, we performed a single ProteinMPNN prediction for each generated protein, using the output of the first run to represent the inferred sequence. This approach was chosen to accelerate the inference process of the model and to ensure that the final perplexity metric is not artificially inflated.

We conduct a comprehensive comparison of DiMA with a suite of existing pre-trained models for generating proteins of varying sizes. Due to the absence of a reference sample in this experiment, we focus on evaluating protein quality and diversity. The results are presented in the Table 8. DiMA, DPLM, ProtGPT2, and RFdiffusion models demonstrated the strongest performance in protein

Table 7: *Comprehensive set of metrics assessing the quality, distribution matching, diversity and novelty of the generated proteins of existing models and DiMA on SwissProt and AFDBv4-90 datasets.*

**Quality**

| | Model | pLDDT (↑) | Progen ppl (↓) | ESM-2 pppl (↓) | scPpl (↓) | TM-score (↑) | BLAST (↑) |
|---|---|---|---|---|---|---|---|
| SwissProt | Dataset | 80.7 | 6.03 | 5.35 | 1.88 | 0.80 | 100 |
| | Random sequences | 24.8 | 21.91 | 21.53 | 2.77 | 0.33 | 0 |
| | Walk-Jump | 32.4 | 15.47 | 14.72 | 2.41 | 0.35 | 1 |
| | RITA | 43.9 | 14.99 | 13.77 | 2.36 | 0.48 | 28 |
| | proteinGAN | 30.4 | 17.58 | 16.48 | 2.57 | 0.00 | 0 |
| | SeqDesign | 43.1 | 12.78 | 11.89 | 2.35 | 0.41 | 17 |
| | EvoDiff-OADM | 37.1 | 16.42 | 15.77 | 2.44 | 0.42 | 12 |
| | D3PM | 36.7 | 16.83 | 16.52 | 2.36 | 0.48 | 9 |
| | DFM | 37.8 | 16.48 | 15.25 | 2.44 | 0.40 | 9 |
| | DPLM | 84.1 | 3.57 | 3.50 | 1.68 | 0.93 | 88 |
| | nanoGPT | 61.0 | 8.87 | 8.18 | 2.04 | 0.63 | 43 |
| | DiMA | 83.3 | 5.07 | 4.68 | 1.17 | 0.87 | 68 |
| AFDB | Dataset | 83.9 | 10.83 | 5.79 | 1.75 | 0.92 | 100 |
| | Random sequences | 26.2 | 22.16 | 21.67 | 2.75 | 0.35 | 0 |
| | nanoGPT | 68.8 | 9.92 | 8.14 | 1.94 | 0.77 | 40 |
| | DPLM | 86.6 | 4.73 | 3.81 | 2.02 | 0.94 | 62 |
| | DiMA | 71.5 | 11.57 | 8.97 | 1.90 | 0.85 | 48 |

**Distributional Similarity**

| | Model | FD-seq (↓) | MMD-seq (↓) | OT-seq (↓) | FD-struct (↓) | MMD-struct (↓) | OT-struct (↓) |
|---|---|---|---|---|---|---|---|
| SwissProt | Dataset | 0.13 | 0.00 | 1.08 | 0.00 | 0.00 | 0.05 |
| | Random sequences | 3.97 | 0.20 | 3.88 | 1.23 | 0.41 | 1.31 |
| | Walk-Jump | 2.63 | 0.33 | 3.56 | 0.61 | 0.05 | 0.69 |
| | RITA | 1.19 | 0.14 | 2.28 | 0.37 | 0.03 | 0.52 |
| | proteinGAN | 2.94 | 0.17 | 3.98 | 0.93 | 0.34 | 1.02 |
| | SeqDesign | 3.53 | 0.19 | 5.12 | 0.95 | 0.25 | 1.11 |
| | EvoDiff-OADM | 1.49 | 0.11 | 2.63 | 0.52 | 0.20 | 0.66 |
| | D3PM | 1.50 | 0.19 | 2.56 | 0.57 | 0.05 | 0.72 |
| | DFM | 1.46 | 0.19 | 2.49 | 0.52 | 0.04 | 0.68 |
| | DPLM | 0.50 | 0.02 | 3.50 | 1.68 | 0.93 | 0.88 |
| | nanoGPT | 1.24 | 0.06 | 2.53 | 0.15 | 0.04 | 0.26 |
| | DiMA | 0.34 | 0.02 | 1.41 | 0.06 | 0.01 | 0.12 |
| AFDB | Dataset | 0.11 | 0.001 | 1.153 | 0.00 | 0.000 | 0.05 |
| | Random sequences | 2.55 | 0.339 | 3.411 | 1.48 | 0.133 | 1.55 |
| | nanoGPT | 0.53 | 0.035 | 1.604 | 0.09 | 0.005 | 0.16 |
| | DPLM | 1.46 | 0.115 | 2.464 | 0.05 | 0.005 | 0.10 |
| | DiMA | 0.27 | 0.017 | 1.499 | 0.03 | 0.005 | 0.10 |

**Diversity and Novelty**

| | Model | Rep (↓) | $CD_{0.5}$ (↑) | $CD_{0.95}$ (↑) | $PCD_{0.5}$ (↑) | $NCD_{0.5}$ (↑) | Novelty (↑) |
|---|---|---|---|---|---|---|---|
| SwissProt | Dataset | 0.045 | 1.000 | 0.943 | 0.990 | 0.304 | 25.35 |
| | Random sequences | 0.000 | 1.000 | 1.000 | 1.000 | 0.000 | 85.11 |
| | Walk-Jump | 0.001 | 1.000 | 1.000 | 0.000 | 1.000 | 82.20 |
| | RITA | 0.028 | 0.988 | 0.998 | 0.125 | 0.861 | 60.45 |
| | proteinGAN | 0.042 | 0.955 | 1.000 | 0.000 | 0.955 | 83.57 |
| | SeqDesign | 0.210 | 0.929 | 1.000 | 0.009 | 0.929 | 81.26 |
| | EvoDiff-OADM | 0.006 | 0.986 | 1.000 | 0.058 | 0.929 | 77.61 |
| | D3PM | 0.003 | 0.994 | 1.000 | 0.025 | 0.968 | 78.43 |
| | DFM | 0.004 | 0.996 | 1.000 | 0.048 | 0.947 | 77.27 |
| | DPLM | 0.781 | 0.494 | 0.812 | 0.267 | 0.236 | 11.56 |
| | nanoGPT | 0.228 | 0.900 | 0.994 | 0.226 | 0.679 | 53.77 |
| | DiMA | 0.320 | 0.611 | 0.992 | 0.246 | 0.392 | 35.74 |
| AFDB | Dataset | 0.008 | 0.994 | 1.000 | 0.029 | 0.966 | 57.65 |
| | Random sequences | 0.000 | 1.000 | 1.000 | 0.000 | 1.000 | 84.68 |
| | nanoGPT | 0.024 | 1.000 | 0.986 | 0.037 | 0.953 | 69.20 |
| | DPLM | 0.285 | 0.970 | 0.476 | 0.132 | 0.341 | 51.58 |
| | DiMA | 0.002 | 1.000 | 1.0 | 0.002 | 0.992 | 72.87 |

structural plausibility and foldability assessment. The remaining baselines exhibit significantly lower quality in terms of perplexity and structural plausibility.

Notably, RFDiffusion, trained on structural representations of proteins, exhibits a high degree of protein structural plausibility, potentially attributed to its structural bias. However, RF-Diffusion exhibits a high perplexity value, suggesting a low quality of predicted amino acid sequences and potentially indicating limitations in the performance of ProteinMPNN. While the combined use of these models for protein sequence generation yields promising results, it does not achieve state-of-the-art performance.

Table 8: *Comparison of the DiMA model with established pre-trained large protein models.*

| Encoder | pLDDT (↑) | Progen ppl (↓) | ESM-2 pppl (↓) | scPpl (↓) | Rep (↓) | CD$_{0.5}$ (↑) | CD$_{0.95}$ (↑) |
|---|---|---|---|---|---|---|---|
| Multiflow-21M | 82.8 | 8.67 | 4.87 | **1.00** | 0.181 | 0.990 | 1.000 |
| Chroma-33M | 66.8 | 12.09 | 7.64 | 1.55 | 0.022 | 1.000 | 1.000 |
| RFDiffusion-80M | 76.7 | 12.07 | 8.05 | 1.25 | 0.018 | 1.000 | 1.000 |
| ProtGPT2-738M | 63.0 | 7.79 | 5.70 | 2.20 | 0.096 | 0.998 | 1.000 |
| ProGen2-151M | 46.2 | 12.78 | 11.33 | 2.39 | 0.084 | 0.998 | 1.000 |
| ProGen2-764M | 50.3 | 12.05 | 10.94 | 2.37 | 0.066 | 0.996 | 0.996 |
| ProGen2-2.7B | 52.3 | 11.78 | 10.57 | 2.35 | 0.044 | 0.992 | 0.994 |
| ProGen2-6.4B | 57.2 | 9.71 | 8.67 | 2.26 | 0.087 | 0.976 | 1.000 |
| EvoDiff-38M | 40.2 | 17.46 | 15.61 | 2.53 | 0.005 | 1.000 | 1.000 |
| EvoDiff-640M | 40.5 | 17.35 | 15.38 | 2.52 | 0.000 | 1.000 | 1.000 |
| ProLLAMA-7B | 53.1 | 10.50 | 7.46 | 2.26 | 0.133 | 0.982 | 1.000 |
| RITA-85M | 40.3 | 18.34 | 16.16 | 2.55 | 0.000 | 1.000 | 1.000 |
| RITA-300M | 41.5 | 19.10 | 15.73 | 2.57 | 0.000 | 0.990 | 0.990 |
| RITA-680M | 42.5 | 20.48 | 15.31 | 2.63 | 0.000 | 0.958 | 0.958 |
| RITA-1.2B | 42.6 | 19.39 | 15.22 | 2.64 | 0.000 | 0.966 | 0.966 |
| DPLM-150M | 81.8 | **3.90** | 2.82 | 1.60 | 0.658 | 0.654 | 0.917 |
| DPLM-650M | 81.8 | 4.36 | **2.41** | 1.60 | 0.533 | 0.746 | 0.943 |
| DPLM-3B | **83.1** | 4.16 | 2.75 | 1.57 | 0.911 | 0.568 | 0.732 |
| DiMA-33M | **83.3** | 5.07 | 4.68 | 1.70 | 0.320 | 0.611 | 0.992 |

Table 9: *Evaluation of the diffusion models utilizing two variants of CHEAP encoders and trained with hyperparameters identified through our ablation studies.*

| CHEAP Encoder | Generation steps | FD-seq (↓) | MMD-seq (↓) | pLDDT (↑) | Progen ppl (↓) | Rep (↓) | CD$_{0.5}$ (↑) | Novelty (↑) |
|---|---|---|---|---|---|---|---|---|
| CHEAP_shorten_1_dim_64 | 250 | **0.304** | **0.0154** | 78.95 | 7.76 | **0.040** | **0.626** | **51.69** |
| | 1000 | 0.322 | 0.0165 | 80.28 | 7.05 | 0.053 | 0.572 | 49.39 |
| | 2000 | **0.309** | 0.0162 | 81.68 | **6.73** | 0.049 | 0.557 | 49.02 |
| CHEAP_shorten_2_dim_64 | 1000 | 0.364 | 0.0203 | 81.38 | 7.14 | 0.041 | 0.561 | 50.74 |
| | 2000 | 0.373 | 0.0206 | **82.00** | 6.94 | 0.047 | 0.541 | 50.04 |

The family of DPLM models demonstrates high protein foldability quality with low perplexity scores, indicating the generation of high-quality proteins. However, DPLM models exhibit a significant drawback in terms of diversity (see Table 8). A substantial portion of subsequences are repeated across multiple generated proteins, negatively impacting the representativeness of the generated proteins. Notably, DPLM-3B generates 27% duplicate sequences, highlighting the challenge of balancing quality with diversity in this model.

DiMA is capable of generating high-quality and diverse protein sequences with reasonable predicted structures. Using 100 times fewer parameters, it achieves comparable quality to other models, like DPLM models, while surpassing them in the diversity of proteins generated. These findings highlight DiMA's potential as a promising approach for protein sequence generation. It balances computational efficiency with the generation of diverse and high-quality proteins.

## C.5 EXPLORATION OF CHEAP ENCODER

In this section, we present an evaluation of the proposed diffusion model employing the CHEAP encoder, utilizing the optimal hyperparameters derived from our comprehensive ablation studies. Our analysis focuses on the performance of the diffusion model across two distinct encoder configurations, one of which incorporates a reduced number of sequence tokens, enabling a more efficient diffusion training. Furthermore, we investigate the influence of varying the number of generation steps on the result quality of the generated proteins. The findings from our experiments reveal that, across both encoder variants, the diffusion model has effectively learned to generate proteins characterized by high quality and substantial diversity. The results of the evaluation are presented in Table C.5.

## C.6 GENERATION OF SEQUENCES FROM SPECIFIC PROTEIN FAMILIES

One widely used approach to generating family-specific proteins is either to train a model from scratch or fine-tune a pre-trained model on a set of similar proteins. We trained and fine-tuned DiMA, nanoGPT, and EvoDiff on seven protein family data. We used the same hyperparameters, parameter counts, and architectures as in training models for unconditional generation.

Table 10: *Quality of generation in terms of pLDDT for models trained from scratch and fine-tuned on various protein families. Higher values correspond to higher structure quality. Model names with "ft" refer to finetuning of a SwissProt version of the particular model. Model names without "ft" refer to training from scratch.*

| Model | LexA | CRISPR | NrdR | PHI | PurE | Lysozyme | GH12 |
|---|---|---|---|---|---|---|---|
| Dataset | $87.3 \pm 5.6$ | $87.1 \pm 5.5$ | $78.4 \pm 3.4$ | $91.2 \pm 3.0$ | $87.0 \pm 2.7$ | $84.7 \pm 4.6$ | $87.9 \pm 4.4$ |
| DiMA | $87.9 \pm 3.9$ | $86.4 \pm 6.0$ | $78.5 \pm 4.3$ | $90.3 \pm 2.7$ | $87.2 \pm 2.4$ | $85.4 \pm 4.0$ | $83.9 \pm 13.1$ |
| DiMA ft | $87.6 \pm 4.5$ | $87.0 \pm 4.4$ | $79.0 \pm 3.0$ | $91.2 \pm 2.3$ | $87.3 \pm 2.2$ | $85.5 \pm 3.8$ | $87.2 \pm 4.3$ |
| nanoGPT | $87.9 \pm 3.6$ | $84.4 \pm 9.2$ | $79.0 \pm 3.7$ | $90.4 \pm 3.1$ | $87.3 \pm 2.0$ | $83.8 \pm 8.2$ | $82.3 \pm 9.5$ |
| nanoGPT ft | $82.3 \pm 11.0$ | $58.9 \pm 18.4$ | $77.9 \pm 3.9$ | $82.1 \pm 12.0$ | $85.4 \pm 5.9$ | $58.6 \pm 16.8$ | $53.4 \pm 19.6$ |
| EvoDiff | $87.1 \pm 5.9$ | $84.7 \pm 7.9$ | $78.7 \pm 3.5$ | $90.2 \pm 4.7$ | $87.0 \pm 2.9$ | $80.7 \pm 10.4$ | $86.1 \pm 7.1$ |
| EvoDiff ft | $87.1 \pm 5.6$ | $86.4 \pm 5.9$ | $78.9 \pm 3.2$ | $90.3 \pm 4.1$ | $87.2 \pm 2.2$ | $82.3 \pm 7.8$ | $86.8 \pm 5.8$ |

Table 11: *Distribution similarity between dataset and generated sequences in terms of Frechet distances on ProtT5 embeddings for models trained from scratch and fine-tuned on various protein families. Smaller values correspond to more similar distributions.*

| Model | LexA | CRISPR | NrdR | PHI | PurE | Lysozyme | GH12 |
|---|---|---|---|---|---|---|---|
| Dataset | 0.013 | 0.012 | 0.012 | 0.021 | 0.016 | 0.028 | 0.012 |
| DiMA | **0.044** | 0.047 | 0.038 | 0.065 | 0.033 | **0.051** | 0.153 |
| DiMA ft | 0.050 | **0.037** | **0.027** | **0.050** | 0.036 | 0.074 | 0.072 |
| nanoGPT | 0.060 | 0.048 | 0.032 | 0.087 | 0.038 | 0.113 | 0.076 |
| nanoGPT ft | 0.183 | 0.489 | 0.115 | 0.251 | 0.049 | 0.744 | 0.930 |
| EvoDiff | 0.047 | 0.040 | 0.049 | 0.075 | **0.028** | 0.102 | **0.045** |
| EvoDiff ft | 0.066 | 0.048 | 0.061 | 0.051 | 0.037 | 0.064 | **0.045** |

To evaluate the models we used pLDDT to measure of quality, FD-seq to assess distribution similarity and BLAST to check if models simply remember sequences from the dataset or generate new ones.

For most protein families, fine-tuning the models led to improved pLDDT scores and reduced FD-seq values compared to training from scratch. The results of evaluation are presented in Tables 10, 11.

## C.7 INPAINTING

To demonstrate that solid performance in regime of unconditional generation enables effective conditional generation we conduct a proof-of-concept experiment on sequence inpainting, a challenging conditional generation task that requires generating novel sequences that maintain structural and functional coherence.

We evaluate DiMA's conditional generation capabilities using 180 sequences from our SwissProt test set, specifically selecting sequences with at most 50% identity to their nearest neighbors in the training set to prevent memorization effects. For each sequence, we mask a random region of variable length (8-50 amino acids) and assess generation quality using multiple stringent criteria: the complete sequence must achieve pLDDT $\geq 80$, the inpainted region must maintain pLDDT $\geq 80$, and the unmasked regions must preserve their structure (RMSD $\leq 1$Å compared to the reference structure). We want to point out, that these criteria are extremely tough, considering that we essentially use language models with no use of 3D-structure data.

To enable conditional generation, we augment DiMA with a lightweight adapter consisting of three transformer blocks, whose outputs are added to all diffusion transformer blocks. This adapter is trained on our unconditional training set with random region masking for 10,000 steps. To account for generation stochasticity, we perform 10 generation attempts per sequence and consider generation successful if any attempt satisfied all quality criteria.

The results demonstrate DiMA's strong performance in conditional generation. DiMA achieves a 42.2% success rate, outperforming both DPLM (40.0%) and random baseline (21.1%). Notably, DiMA generates inpainted regions with substantially higher average quality (pLDDT 66.9) compared to DPLM (59.3) and random baseline (50.9). Furthermore, the generated sequences show significant

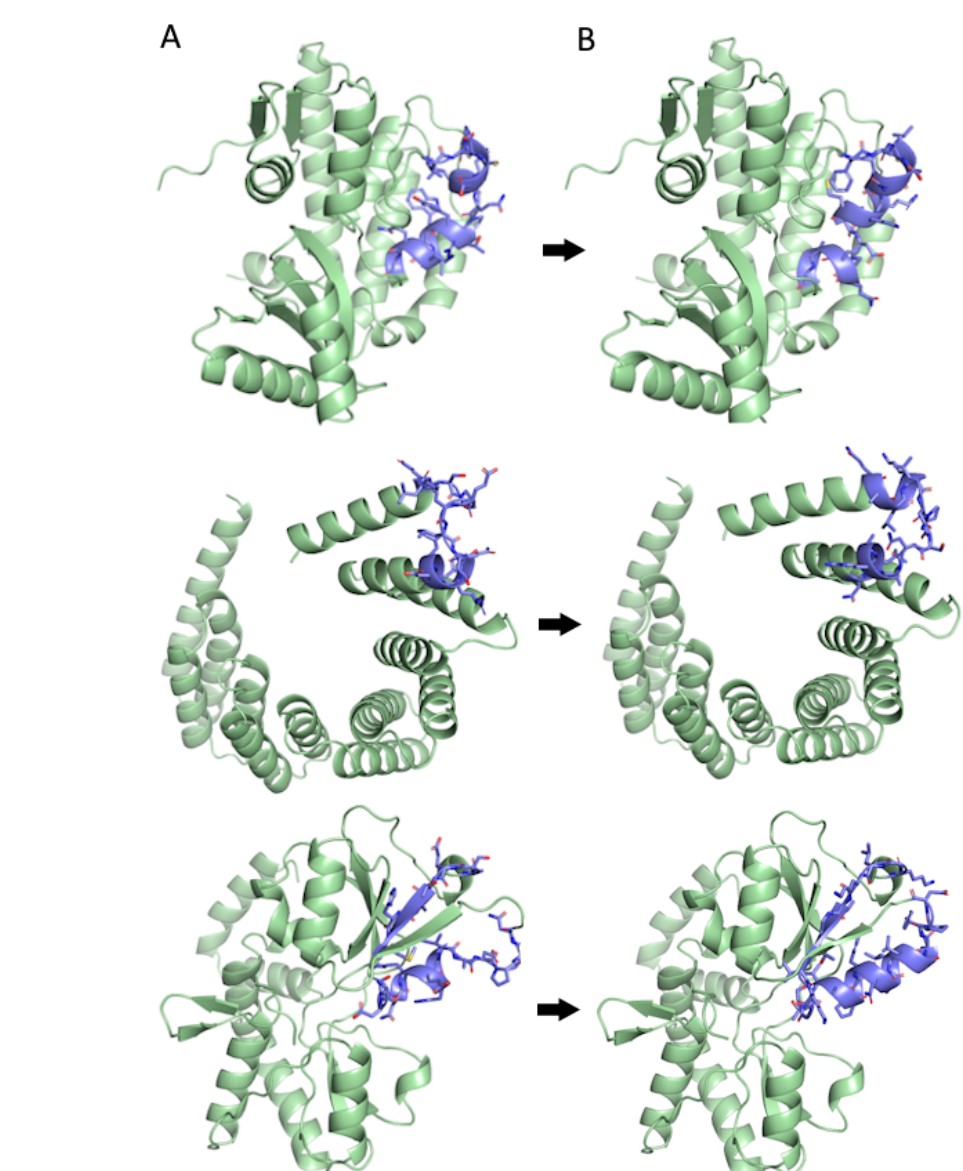

Figure 8: Inpainting example generations. A- reference proteins, B- DiMA generated proteins. DiMA can produce different inpaint region, conditioned on other parts.

novelty (inpainted region average novelty of 80), indicating that DiMA is not simply memorizing training data but generating novel, structurally plausible sequences.

Figure 8 depicts the examples of successfully inpainted regions. These results clearly demonstrate that DiMA can be effectively adapted for conditional generation tasks through established mechanisms like adapter-based conditioning.

## D  BIOLOGICAL RELEVANCE ANALYSIS

**Superfamily annotation**. For proteins annotation we utilized the established protein annotation tool InterProScan (Paysan-Lafosse et al., 2023; Jones et al., 2014). InterProScan includes a set of pre-trained models based on hidden Markov models (HMMs), which allow for assigning potential folds and functions. This analysis involves annotating the generated protein sequences using the

Table 12: *Performance comparison of DiMA and DPLM on the inpainting task, measured by three metrics: success rate, average quality (Region pLDDT), and average novelty (Region identity).*

| Model | Success rate, % (↑) | Region pLDDT (↑) | Region Novelty (↑) |
|---|---|---|---|
| DiMA | 42.2 | 66.9 | 80 |
| DPLM | 40.0 | 59.3 | 75 |
| Random | 21.1 | 50.9 | 92 |

Figure 9: Histogram depicting the occurrence of the top 15 most frequent SUPERFAMILY domains in the SwissProt dataset pool. (Oates et al., 2015; Jones et al., 2014). x- Fraction of each annotation per model.

SUPERFAMILY HMM library (Oates et al., 2015), which provides sequence homology to SCOP structural domains (Murzin et al., 1995). **IDR exploration**. Natural proteins encompass both structured regions and IDRs that lack regular structure but still play functional roles (Uversky, 2015) (Figure 12). To annotate these regions, we employ the MobiDB model within the InterProScan tool, which predicts IDRs in protein sequences using multiple classifiers (Piovesan et al., 2018). Sequences generated by DiMA exhibit a natural-like profile of IDR length distribution (Figure 12). Generation of both folded and unfolded structural regions provides a distinct advantage for sequence diffusion models over models exclusively trained on folded protein domains. **Secondary structure exploration**. Finally, we calculate the frequency of secondary structure elements within the folded regions using the DSSP tool (Kabsch & Sander, 1983) against protein structures predicted via ESMFold. DiMA mirrors the amount of secondary structural elements of natural proteins. DiMA generates seuqnces with number of secondary elements close to relevant number in validation dataset (Figure 13).

# E    MODEL DETAILS

## E.1    MODEL ARCHITECTURE

We employ a 12-layer Transformer model with 16 attention heads and a hidden size of 320 as the backbone for our diffusion model, incorporating several modifications specifically designed to optimize the denoising diffusion process in the context of protein-related data. To enhance the model's performance, we first introduce trainable positional encodings to the noisy protein latents, allowing the model to better capture the sequential nature of the data. The input for each transformer block

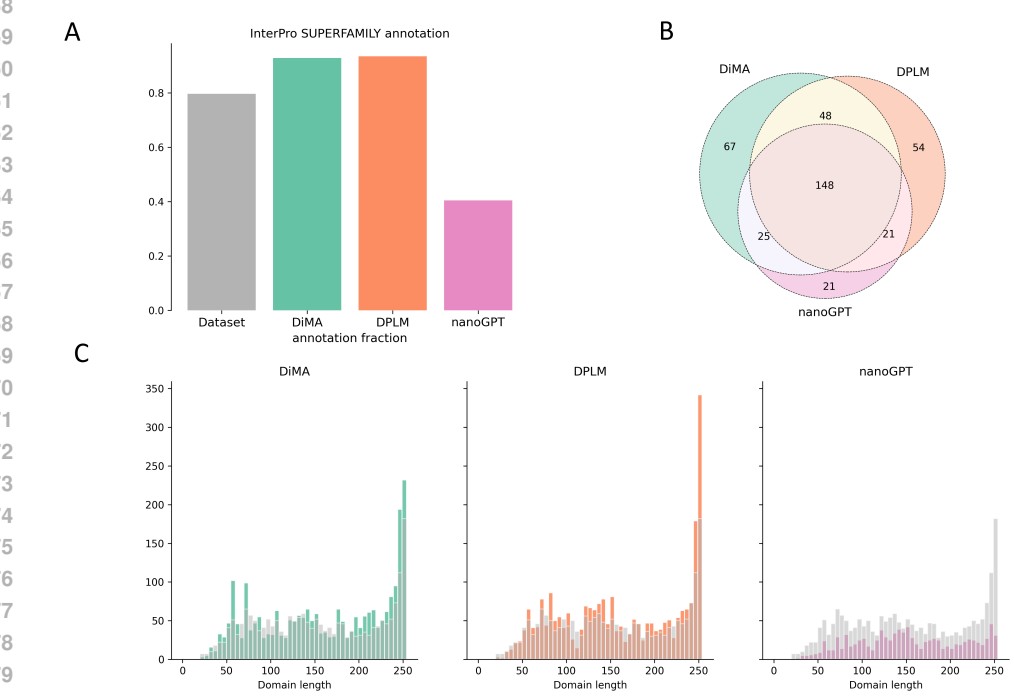

Figure 10: Sequence annotation into known structural domains using SUPERFAMILY tool within InterProScan (Oates et al., 2015; Jones et al., 2014). Histogram of domain lengths (Grey - 2048 dataset sequences; colored - 2048 generated sequences).

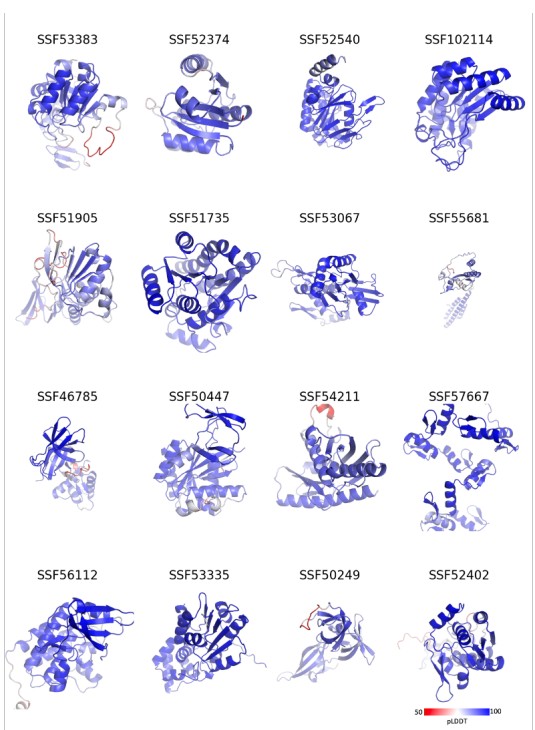

Figure 11: Sequence annotation into known structural domains using SUPERFAMILY tool within InterProScan (Oates et al., 2015; Jones et al., 2014). ESMFold-predicted structures of representative SUPERFAMILY domains generated by DiMA.

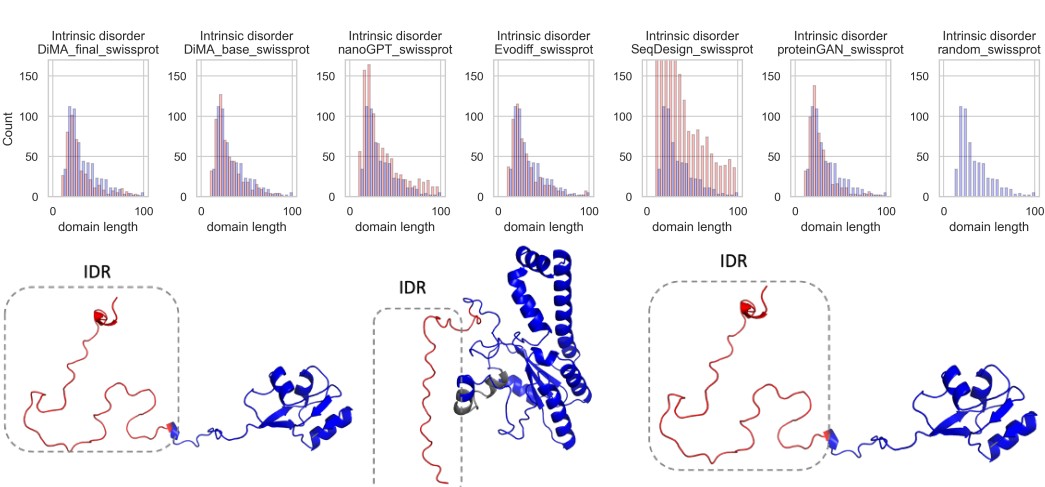

Figure 12: Prediction of intrinsic disorder regions (IDR) using the MobiDBLite tool (Piovesan et al., 2018). (A) Histogram depicting the lengths of intrinsic disorder regions. The blue color represents the dataset, while the red color represents the generated sequences. No hits were found for random sequences. (B) Representative examples of proteins generated by DiMA, highlighting intrinsic disorder regions in red and folded structural domains in blue.

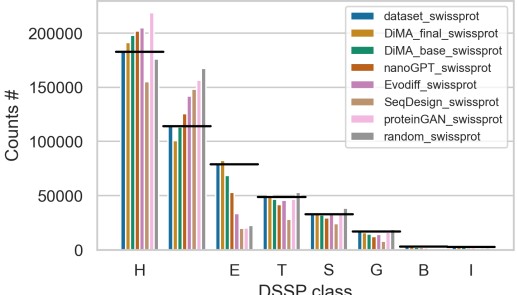

Figure 13: The number of secondary structure elements calculated per residue from ESMFold predicted structures using DSSP (Kabsch & Sander, 1983) software. H = $\alpha$-helix; B = residue in isolated $\beta$-bridge; E = extended strand, participates in $\beta$ ladder; G = 3-helix (310 helix); I = 5 helix ($\pi$-helix); T = hydrogen bonded turn; S = bend.

is constructed as a sum of the output from the previous block, time embeddings, and self-condition predictions, which are projected through linear layers. This approach facilitates the integration of temporal information and improves the model's ability to learn complex patterns. Additionally, we implement long skip connections, recognizing that for time steps close to zero, the model's output closely resembles the input. This modification is crucial as it aids in learning an identity transformation, thereby stabilizing the training process and enhancing the model's overall efficacy. The architecture of our model is illustrated in Figure 14.

## E.2 TRAINING DETAILS

All models were trained with a batch size of $512$ and a learning rate of $1e^{-4}$ to convergence. We clip our gradient norm to $2$ and have a linear warmup schedule for the first $5000$ iterations. We also use a 0.9999 EMA.

The experiments were conducted using $4$ A100 80GB GPUs. Each training session lasts approximately 10 days

## E.3 LENGTH SAMPLING

During the inference phase, the model needs to define the length of the generated sequence. We compare two approaches to tackle this problem: training diffusion models both with and without pad masking. In the first case, we feed additionally to corrupted latents the attention mask of pad tokens during training and ignore pad tokens for computing diffusion loss. During inference, we sample the length from the empirical distribution of lengths in the training set. In the second case, we do not provide any information about pad tokens during training and compute loss using all tokens in sequence, including pad. Then, during generation, the model should define the length by itself. Figure 15 depicts the distribution of lengths in the training and generated by second approach sets. The distribution of generated sequences differs from the training set on both datasets. To avoid this distribution mismatch, we use an attention mask during training and sample length during inference.

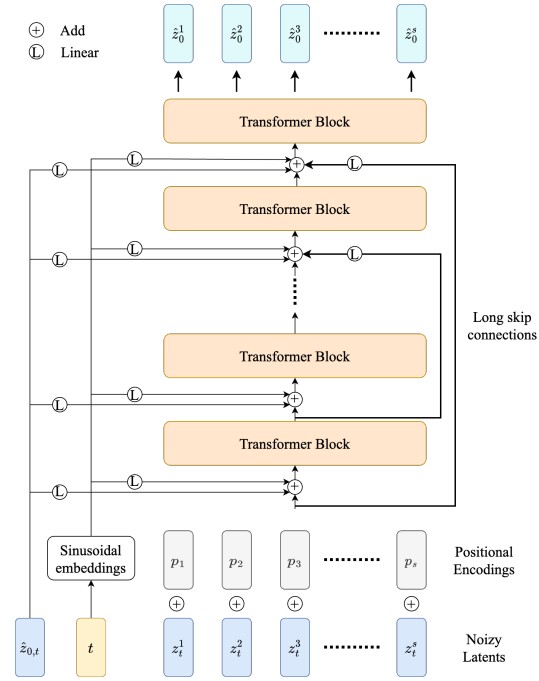

Figure 14: The architecture of the denosing model.

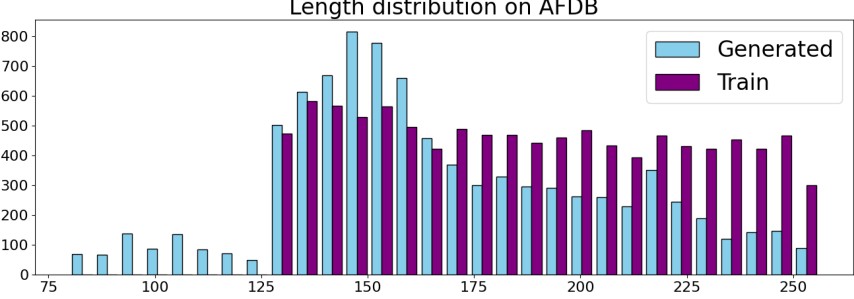

Figure 15: The distribution of lengths in the training and generated sets for models trained on AFDB.

