# OpenReview forum: "Diffusion on language model encodings for protein sequence generation"
_ICLR.cc/2025/Conference — Submitted to ICLR 2025_

### Official Review · Reviewer_fqCm · 2024-10-28

**Soundness:** 2
**Presentation:** 2
**Contribution:** 2
**Rating:** 3
**Confidence:** 5

**Summary:**

The paper presents DiMA (Diffusion on Language Model Encodings for Protein Sequence Generation), a framework for generating protein sequences. DiMA addresses the challenge of generating high-quality, diverse protein sequences by leveraging latent diffusion on protein language model (pLM) encodings from ESM-2. Existing protein generation models struggle to balance quality, diversity, and parameter efficiency, especially for unconditional generation, which is essential for creating novel protein structures without predefined constraints.

**Strengths:**

1. The combination of latent diffusion modeling with protein language model encodings is interesting.
2. The manuscript is well-organized and clearly written.
3. The experiments are extensive, with a set of rigorous evaluation metrics.

**Weaknesses:**

1. While latent diffusion is novel in this context, the motivation for choosing this approach over more conventional methods like sequence-based discrete diffusion or 3D-structure-based continuous diffusion is not strongly justified. The authors could strengthen their argument by highlighting specific challenges with current discrete or continuous approaches in handling the protein data structure.
2. It omits comparisons with strong baseline models like Chroma [1] and MultiFlow [2].
3. Although DiMA is designed to generate diverse sequences, the paper does not sufficiently analyze sequence variability at the biochemical or functional level.
4. The current citation format, as mentioned, should follow the convention of \citep rather than \cite for clarity and consistency.


[1] Ingraham, John B., Max Baranov, Zak Costello, Karl W. Barber, Wujie Wang, Ahmed Ismail, Vincent Frappier et al. "Illuminating protein space with a programmable generative model." Nature 623, no. 7989 (2023): 1070-1078.

[2] Campbell, Andrew, Jason Yim, Regina Barzilay, Tom Rainforth, and Tommi Jaakkola. "Generative Flows on Discrete State-Spaces: Enabling Multimodal Flows with Applications to Protein Co-Design." In Forty-first International Conference on Machine Learning.

**Questions:**

The paper is a valuable contribution to protein sequence generation through latent diffusion modeling. However, the work could be strengthened by deeper motivation for the latent diffusion choice and comparisons with recent models like Chroma and MultiFlow. I really admire this work for its comprehensive evaluation, but I doubt it is good enough for this top AI conference.

---

> ### Comment · Reviewer_fqCm · 2024-11-26
>
> I am more than willing to engage with the rebuttal; however, the author has not yet provided a response at this time.

---

> ### Author Response · Authors · 2024-11-27
>
> Dear Reviewer,
>
> Thank you for your thoughtful and constructive feedback on our manuscript. Your comments have helped us identify important areas for improvement, and we appreciate the time you took to provide detailed suggestions. Let us address each of your points.
>
> >**W1**. While latent diffusion is novel in this context, the motivation for choosing this approach over more conventional methods like sequence-based discrete diffusion or 3D-structure-based continuous diffusion is not strongly justified. The authors could strengthen their argument by highlighting specific challenges with current discrete or continuous approaches in handling the protein data structure.
>
> >**Q1**. the work could be strengthened by deeper motivation for the latent diffusion choice.
>
> Our approach offers several key advantages that we should have articulated more clearly in the manuscript. First, the continuous nature of the latent space enables direct application of established score-based techniques like classifier and classifier-free guidance without requiring discrete approximations. This creates opportunities for more controlled and directed protein generation. Second, training in continuous latent spaces is generally more stable and efficient compared to discrete spaces, as demonstrated by recent work comparing continuous vs discrete representations (e.g., CHEAP [1]). Finally, as shown in our ablation studies, the continuous latent space allows fine-grained optimization of multiple aspects of the diffusion process - from noise scheduling to architectural choices. We have updated our manuscript, to reflect this motivation.
>
> >**W2**. It omits comparisons with strong baseline models like Chroma [1] and MultiFlow [2].
>
> >**Q1**. the work could be strengthened by comparisons with recent models like Chroma and MultiFlow.
>
> We sincerely thank you for bringing up these strong baselines. We have now included comprehensive comparisons with both models. To ensure fair comparison, we evaluated sequences extracted from their generated PDB files using the same metrics applied to other baselines. Our updated results show that MultiFlow achieves impressive performance (pLDDT: 82.8, scPpl: 1.0, $CD_{0.5}$: 0.99), while Chroma demonstrates solid results (pLDDT: 66.8, scPpl: 7.64, $CD_{0.5}$: 1.0). On the quality-diversity trade-off scale MultiFlow is tilted towards the diversity. However, the mechanism of sequence-structure co-generation in both MultiFlow and Chroma leads to outstanding results in scPerplexity, which evaluates the self-consistensy between the two modalities. Surprisingly, these results highlight an interesting advantage of our approach - its applicability to **any** continuous latent space. For example, following the suggestion of the Reviewer w8qc, we experimented with CHEAP latents that enable both sequence and structure decoding, achieving strong results (pLDDT: 81.7, FD-seq: 0.309) without any architectural modifications.
>
> | Model | pLDDT | Progen ppl | ESM-2 pppl | scPpl | Rep | CD_{0.5} | CD_{0.95} |
> | --- | --- | --- | --- | --- | --- | --- | --- |
> | Chroma 33M | 66.8 | 12.09 | 7.64 | 1.55 | 0.02 | 1.0 | 1.0 |
> | MultiFlow 21M | 82.8 | 8.67 | 4.87 | 1.00 | 0.18 | 0.99 | 1.0 |
> | DPLM 3B | 83.1 | 4.16 | 2.75 | 1.6 | 0.911 | 0.568 | 0.732 |
> | DiMA 35M | 83.3 | 5.07 | 4.68 | 1.7 | 0.320 | 0.611 | 0.992 |
>
> We have included the results on all evaluations in the revised version of the manuscript.
>
> >**W3**. Although DiMA is designed to generate diverse sequences, the paper does not sufficiently analyze sequence variability at the biochemical or functional level.
>
> We appreciate you highlighting this aspect of our work. While we did conduct the biological and functional analysis, we acknowledge we could have presented it more clearly. Our biological relevance section demonstrates that DiMA successfully captures the functional diversity of natural proteins. Using InterProScan annotation, we show that 92% of DiMA-generated sequences are annotated as known SUPERFAMILY members, closely matching the dataset's 80% annotation rate. Importantly, DiMA accurately reproduces the natural distribution of domain lengths, with most domains spanning 50-75 amino acids. Additionally, our analysis shows DiMA effectively models both structured regions and intrinsically disordered regions (IDRs), capturing the full spectrum of protein functionality. We have reorganized these results in Section 4 and Appendix E for better clarity.
>
> >**W4**. The current citation format, as mentioned, should follow the convention of \citep rather than \cite for clarity and consistency.
>
> Thank you for catching this inconsistency. We have updated all citations to use the proper \citep format.
>
> ---
>
> We thank you for your feedback and suggestions. We would be grateful if you would consider raising your score, in case we have addressed your concerns. Please let us know if any questions still need clarification.

---

> > ### Author Response · Authors · 2024-12-03
> >
> > Dear Reviewer fqCm,
> >
> > Once again, thank you for your time and thoughtful feedback. Considering the approaching end of the rebuttal period, we would like to answer any further questions if any.
> >
> > Best regards,
> > Authors

---

### Official Review · Reviewer_PQDj · 2024-11-02

**Soundness:** 3
**Presentation:** 3
**Contribution:** 3
**Rating:** 6
**Confidence:** 3

**Summary:**

The paper introduces DiMA, a latent diffusion model designed for generating protein sequences using embeddings from the pre-trained protein language model ESM-2. DiMA aims to address the limitations of current models, which often struggle to achieve both high quality and diversity in unconditional protein sequence generation.

This work provides an in-depth exploration of model components and performs extensive evaluation across multiple metrics to validate DiMA’s sequence quality and diversity.

**Strengths:**

- **Comprehensive Evaluation Metrics**: The authors use a robust set of evaluation metrics, covering quality, distribution similarity, diversity, biological relevance, and both sequence-based and structure-based evaluations.

- **Well-Designed Ablation Study**: The study effectively demonstrates the contribution of each component, particularly the ESM encoder and self-conditioning. The authors also investigate the impact of the noise schedule and model scaling, offering valuable insights into best practices for latent diffusion models in protein generation.

- **Empirical Performance**: The authors compare DiMA against a broad range of baseline models and pretrained models for the unconditional generation task, providing a thorough analysis. DiMA achieves comparable generation quality with larger pretrained models using significantly fewer parameters.

**Weaknesses:**

- **Limited to Unconditional Generation**: DiMA’s design is currently limited to unconditional generation, which poses challenges for performing tasks such as sequence inpainting, motif scaffolding, or guided design. This is a key limitation of DiMA and diffusion models in continuous embedding space. I suggest that the authors discuss and explore the potential of extending DiMA to these tasks, which could greatly enhance the versatility of this work.

- The authors could also consider evaluating DiMA on protein representation learning tasks, similar to those presented in DPLM.

- In the main experiment, it is stated, “For a fair comparison, we train each method from scratch with the same parameter count (33M) as DiMA on the same dataset.” However, DiMA’s dependency on the ESM-2 encoder increases the effective parameter count, which could affect the reliability of these comparisons. This should be clarified to ensure fair evaluation.

- **Inconsistencies in Tables**: Some tables lack consistency, particularly Table 5, where the best-performing models are not consistently bolded. For example, in the Progen ppl column, DPLM-3B is bolded, even though DPLM-150M performs better. Correcting these inconsistencies would improve clarity and fairness.

- **Paper Structure**: Improving the structure of the paper would enhance readability. For example, Section 4 currently only has one subsection (4.1). Creating subsections for ablation studies and other experiments could provide better organization and make the results easier to follow.

**Questions:**

- Please review and correct the citation formatting for consistency.

- I suggest including the forward transition kernel in Section 3, clarifying how the noise schedule is used to determine the mean and variance.

- For the CD0.95 metric, does a higher value indicate greater diversity? In Table 2, 0.990 is bolded, suggesting it is considered the best. Please clarify this metric.

- I wonder why is the FD-seq metric not zero for the ground truth dataset, Is this due to sampling variability? Additionally, FD-STRUCT appears consistently lower than FD-seq—is this due to the lower dimensionality of the structure representation?

- What was the intuition behind choosing 320 as the hidden dimension for the diffusion backbone model? Also, have you considered using the fine-tuned ESM-2 model directly, as seen in DPLM?

- **Potential Bias in pLDDT Evaluation**: Since you use ESMFold to evaluate pLDDT and the ESM-2 encoder in DiMA, could this create a risk of biased evaluation? It may be worth considering alternative pLDDT metrics, such as those from AF2 or OmegaFold 2, to mitigate potential bias.

---

> ### Author Response · Authors · 2024-11-27
>
> Dear Reviewer,
> Thank you for your thorough and constructive review. We greatly appreciate your recognition of our comprehensive evaluation framework, well-designed ablation studies, and empirical performance analysis. We have carefully considered your concerns and suggestions, and we would like to address each point:
>
> >**W1**. Limited to Unconditional Generation: DiMA’s design is currently limited to unconditional generation, which poses challenges for performing tasks such as sequence inpainting, motif scaffolding, or guided design. This is a key limitation of DiMA and diffusion models in continuous embedding space. I suggest that the authors discuss and explore the potential of extending DiMA to these tasks, which could greatly enhance the versatility of this work.
>
> Thank you for raising the important point about conditional generation capabilities. While our work indeed focuses on unconditional generation, we want to demonstrate that this foundation enables effective conditional generation. We have conducted **a proof-of-concept experiment on sequence inpainting**, a challenging conditional generation task.
>
> We evaluate DiMA's conditional generation capabilities using 180 sequences from our SwissProt test set, specifically selecting sequences with at most 50% identity to their nearest neighbors in the training set to prevent memorization effects. For each sequence, we mask a random region of variable length (8-50 amino acids) and assess generation quality using multiple stringent criteria: the complete sequence must achieve pLDDT ≥ 80, the inpainted region must maintain pLDDT ≥ 80, and the unmasked regions must preserve their structure (RMSD ≤ 1Å compared to the reference structure). We want to point out, that these **criteria are extremely tough**, considering that we essentially use language models with no use of 3D-structure data.
>
> To enable conditional generation, we augment DiMA with a lightweight adapter consisting of three transformer blocks, whose outputs are added to all diffusion transformer blocks. This adapter is trained on our unconditional training set with random region masking for 10,000 steps. To account for generation stochasticity, we perform 10 generation attempts per sequence and consider generation successful if any attempt satisfied all quality criteria.
>
> The results demonstrate DiMA's strong performance in conditional generation. DiMA achieves a 42.2% success rate, outperforming both DPLM (40.0%) and random baseline (21.1%). Notably, DiMA generates inpainted regions with substantially higher average quality (pLDDT 66.9) compared to DPLM (59.3) and random baseline (50.9). Furthermore, the **generated sequences show significant novelty** (inpainted region average novelty of 80%), indicating that DiMA is not simply memorizing training data but generating novel, structurally plausible sequences.
>
> These results clearly demonstrate that DiMA can be effectively adapted for conditional generation tasks through established mechanisms like adapter-based conditioning. While our paper focuses on establishing strong unconditional generation capabilities, these experiments show that DiMA provides a robust foundation for various conditional generation tasks.
>
> >**W2**. The authors could also consider evaluating DiMA on protein representation learning tasks, similar to those presented in DPLM.
>
> We appreciate the reviewer's suggestion regarding representation learning tasks. However, it's important to note that DPLM and DiMA fundamentally differ in how they process and generate features. DPLM can be viewed as a specialized version of ESM-2 trained on masked language modeling, making its final layer features naturally suitable for representation learning tasks. In contrast, DiMA is explicitly trained to reconstruct the representations provided by the ESM-2 encoder, making direct application of its features to representation learning tasks conceptually problematic.
>
> Nevertheless, we see promising potential for adapting DiMA's features for representation learning by drawing inspiration from recent advances in the image domain (e.g. https://arxiv.org/pdf/2112.03126). For instance, we could investigate feature aggregation across different diffusion timesteps $t$ and analyze their predictive power for tasks such as DeepLoc or secondary structure prediction. This would involve identifying which timesteps yield the most informative features for different prediction tasks.
>
> This direction for future study represents a substantial research in its own right, requiring careful experimental design and extensive evaluation. Such an undertaking falls outside the timeframe of this rebuttal. We intend to pursue this research direction in future work and believe it could yield valuable insights into how diffusion models learn and represent protein properties across different denoising steps.

---

> ### Author Response · Authors · 2024-11-27
>
> >**W3**. In the main experiment, it is stated, “For a fair comparison, we train each method from scratch with the same parameter count (33M) as DiMA on the same dataset.” However, DiMA’s dependency on the ESM-2 encoder increases the effective parameter count, which could affect the reliability of these comparisons. This should be clarified to ensure fair evaluation.
>
> We want to clarify that our comparison is indeed fair because **we do not use the ESM-2 encoder during inference**. The encoder is only used during training to obtain representations, making the effective parameter count during deployment equal for all compared models.
>
> >**W4**. Inconsistencies in Tables: Some tables lack consistency, particularly Table 5, where the best-performing models are not consistently bolded. For example, in the Progen ppl column, DPLM-3B is bolded, even though DPLM-150M performs better. Correcting these inconsistencies would improve clarity and fairness.
>
> Thank you for noting the inconsistencies in the tables. We have revised the manuscript to ensure consistent highlighting of the best-performing models across all metrics.
>
> >**W5**. Paper Structure: Improving the structure of the paper would enhance readability. For example, Section 4 currently only has one subsection (4.1). Creating subsections for ablation studies and other experiments could provide better organization and make the results easier to follow.
>
> We appreciate your feedback on the paper structure and have reorganized the manuscript, including **Section 4**, into clear thematic subsections to improve readability.
>
> >**Q1**. Please review and correct the citation formatting for consistency.
>
> We have corrected the citation formatting according to the guidelines throughout the text.
>
> >**Q2**. I suggest including the forward transition kernel in Section 3, clarifying how the noise schedule is used to determine the mean and variance.
>
> We have revised **Section 3** of the manuscript for more clarify. The forward transition kernel equation:  $z_t = \sqrt{\alpha_t} z_0 + \sqrt{1 - \alpha_t} \varepsilon$
>
> >**Q3**. For the CD0.95 metric, does a higher value indicate greater diversity? In Table 2, 0.990 is bolded, suggesting it is considered the best. Please clarify this metric.
>
> For this metric, a higher value does indeed indicate greater diversity, as it represents the proportion of unique clusters when sequences are clustered at 95% identity threshold. However, our goal is to generate sequences that match the diversity characteristics of natural proteins. Therefore, we consider the optimal value to be that of the training dataset (0.990 in case of SwissProt), rather than the maximum possible value of 1.0. We have revised our manuscript to make this clearer by adding an explicit explanation in the table captions and metrics description section. Thank you for bringing this to our attention.
>
> >**Q4**. I wonder why is the FD-seq metric not zero for the ground truth dataset, Is this due to sampling variability? Additionally, FD-STRUCT appears consistently lower than FD-seq—is this due to the lower dimensionality of the structure representation?
>
> The distributional metrics for datasets are calculated between two independent samples (2048 sequences each) drawn from the training and hold-out sets. The non-zero values arise from the finite sample size used in the estimation. However, this does not impact the validity of our evaluation framework, as the distributional distance between dataset subsamples remains consistently and significantly smaller than any model's results across all our experiments.
>
> Your observation about FD-struct being consistently lower than FD-seq is astute. This difference likely stems from the dimensionality of the feature spaces - ProtT5 sequence embeddings have dimension 1024, while ProteinMPNN structural representations use dimension 128. However, it's important to note that Fr´echet distance lacks a unified scale, making direct comparisons between different distributional distances problematic. This is one of the reasons why we include both Dataset and Random sequences as reference points in our baseline comparisons - they provide context for interpreting the scale of each metric independently.

---

> ### Author Response · Authors · 2024-11-27
>
> >**Q6**. Potential Bias in pLDDT Evaluation: Since you use ESMFold to evaluate pLDDT and the ESM-2 encoder in DiMA, could this create a risk of biased evaluation? It may be worth considering alternative pLDDT metrics, such as those from AF2 or OmegaFold 2, to mitigate potential bias.
>
> Thank you for raising this important concern about potential evaluation bias. We actually ruled out this issue in the early stages of our work and, thus, did not discuss it in the manuscript. To address your concern directly, we have conducted additional experiments comparing pLDDT scores from both ESMFold and OmegaFold on two batches of 2048 sequences each. Importantly, we included sequences from both DiMA and nanoGPT, where nanoGPT serves as a good control since it has no connection to ESM-2 architecture or training.
>
> The results demonstrate remarkable consistency across both folding models:
> | Model | ESMFold | OmegaFold |
> | --- | --- | --- |
> | DiMA | 80.82 | 81.98 |
> | nanoGPT | 60.97 | 63.57 |
>
> These nearly identical scores from two independent folding models strongly suggest that our evaluation using ESMFold does not introduce significant bias in assessing structural quality. The consistent performance gap between DiMA and nanoGPT is maintained across both evaluation methods, supporting the robustness of our conclusions.
>
> ---
>
> We hope these clarifications address all your concerns. Based on these responses, We would be grateful if you would consider raising your score. If you have any additional questions, we would be glad to address them.

---

> > ### Author Response · Authors · 2024-12-03
> >
> > Dear Reviewer PQDj,
> >
> > We appreciate your time and detailed feedback. Considering the approaching end of the rebuttal period, we would like to ask you if there are any questions left  for us to address.
> >
> > Best regards,
> > Authors

---

### Official Review · Reviewer_GJwR · 2024-11-03

**Soundness:** 2
**Presentation:** 2
**Contribution:** 2
**Rating:** 5
**Confidence:** 4

**Summary:**

The paper presents DiMA, a model leveraging latent diffusion on protein language model encodings (in specific, ESM2) for generating protein sequences. The authors claim that DiMA achieves superior quality, diversity, and distribution matching compared to existing autoregressive and discrete diffusion protein sequence language models while using fewer parameters.

**Strengths:**

- The paper claims extensive evaluations using multiple metrics, which basically supports the findings.
- The idea of doing diffusion or flow matching in the latent space makes sense and is a good direction to explore.
- DiMA’s efficiency in utilizing ten times fewer parameters than leading models is a significant advantage, suggesting potential for broader applications in protein design.

**Weaknesses:**

- Firstly, the approach presented in this paper does not significantly advance the field of **generative models**(the primary area selected). While the application of latent diffusion is noted, it does not bring substantial new insights or methodologies to the existing body of research on generative models (even in the field of protein design). I agree on the experimental contribution of this paper and think the protein design problem is important and relevant. However, it is under-qualified as a top-conference paper given its presentation and methodology contribution.
- Inconsistent results presentation. In Tables 4 and 5 of the main text, the bold digits intended to denote the best results (i guess) are inconsistently labeled. This lack of clarity can lead to confusion about which results are truly the best and detracts from the paper’s credibility.
- The use of Swissprot and AFDB as sequence-based protein databases is not adequately justified. The authors should provide a clearer rationale for selecting these databases and explain how they enhance the effectiveness of the sequence generation process. Why not use Uniprot or Uniref on which ESM2 is pre-trained?
- The authors assert that they use “the most comprehensive set of metrics” to evaluate their model. However, this claim lacks substantiation within the paper, raising questions about the thoroughness of their evaluation process.
- The sections/paragraphs are badly organized and hard to follow, please consider improve the manuscript during rebuttal.

**Questions:**

- Can you clarify how DiMA’s approach contributes uniquely to the generative model area compared to existing methodologies?
- What specific rationale guided your choice of Swissprot and AFDB for protein sequence generation, and how do these structure-based databases specifically benefit your model? If this is the common practice in previous work, do the authors think incorporating sequence-based database like UniProt/UniRef will enhance the performance of DiMA? For example, in one of the baselines, EvoDiff, the authors there use evolutionary-scale sequence data to train their model.
- Could you provide a more detailed breakdown of the metrics used in your evaluations and other studies (eg., baselines) to support your claim of comprehensiveness of evaluation? For example, you can make a table to justify and list the necessity of some newly introduced metrics. That would be helpful for new readers to understand, ok, this is important to evaluate the protein design models and bring new insights to the community.
- Why the authors choose ESM2-8M - a very small representation learning model as the encoder across the study. As a usual practice, most of works use 650M model. Could the author reason the necessity of using 8M model? Moreover, as a control comparison, in Table 3, 10, 11, the authors combine “DiMA-8M” with other ESM2 encoders while in the main experiment, the authors claims using DiMA-33M. How do the authors explain the discrepancy between? Or is this a typo?
- Could the author provide a illustration figure in the main text? That would be helpful for readers to quickly grasp what you have done.

---

> ### Author Response · Authors · 2024-11-27
>
> Thank you for your thorough review and thoughtful feedback. We appreciate you highlighting the strengths of our work, particularly our extensive evaluations across multiple metrics and exploration of the latent diffusion through the ablation studies the ablation studies. We aim to address your concerns and questions below.
>
> >**W1**. Firstly, the approach presented in this paper does not significantly advance the field of **generative models**(the primary area selected). While the application of latent diffusion is noted, it does not bring substantial new insights or methodologies to the existing body of research on generative models (even in the field of protein design). I agree on the experimental contribution of this paper and think the protein design problem is important and relevant. However, it is under-qualified as a top-conference paper given its presentation and methodology contribution.
>
> >**Q1**. Can you clarify how DiMA’s approach contributes uniquely to the generative model area compared to existing methodologies?
>
> While latent continuous diffusion is an established approach, its successful application to protein sequence generation has remained an open challenge. Previous attempts like Pro-LDM [1] are limited to narrow family-specific generation using fixed architectures. Our work represents the first comprehensive study demonstrating that continuous latent diffusion can be effectively adapted for protein sequence generation through careful architectural choices and training procedures. The extensive ablation studies provide insights about components necessary for successful protein generation using this approach. As the reviewer w8qc has suggested, we swapped the ESM-2 encoder with recently developed CHEAP latents that allow both sequence and structure decoding [2]. Without tuning a single architectural knob this setup yields pLDDT 81.7 and FD-seq 0.309, **a very strong result**, second only to our orignal model (**Table 3** of the revised manuscript). This success demonstrates the power of our approach and possibilities that our study provides for the broader community.
>
> >**W3**. The use of Swissprot and AFDB as sequence-based protein databases is not adequately justified. The authors should provide a clearer rationale for selecting these databases and explain how they enhance the effectiveness of the sequence generation process. Why not use Uniprot or Uniref on which ESM2 is pre-trained?
>
> >**Q2**. What specific rationale guided your choice of Swissprot and AFDB for protein sequence generation, and how do these structure-based databases specifically benefit your model? If this is the common practice in previous work, do the authors think incorporating sequence-based database like UniProt/UniRef will enhance the performance of DiMA? For example, in one of the baselines, EvoDiff, the authors there use evolutionary-scale sequence data to train their model.
>
> We would like to clarify that **both datasets are sequence-based**, not structure-based. SwissProt is a manually curated section of UniProtKB containing high-quality sequence data (https://www.uniprot.org/help/uniprotkb_sections). AFDBv4_90 [3] is derived from UniRef50 sequences, filtered to include only those with predicted pLDDT > 90. This filtering ensures that any quality issues in generated sequences stem from model architecture or training, not training data quality. Using both a carefully curated dataset (SwissProt) and a larger filtered dataset (AFDBv4_90) allows us to evaluate model performance across different data regimes.
>
> >**W4**. The authors assert that they use “the most comprehensive set of metrics” to evaluate their model. However, this claim lacks substantiation within the paper, raising questions about the thoroughness of their evaluation process.
>
> >**Q3**. Could you provide a more detailed breakdown of the metrics used in your evaluations and other studies (eg., baselines) to support your claim of comprehensiveness of evaluation? For example, you can make a table to justify and list the necessity of some newly introduced metrics. That would be helpful for new readers to understand, ok, this is important to evaluate the protein design models and bring new insights to the community.
>
> In our evaluation framework, we use three categories of metrics that assess different aspects of protein generation. These are the metrics of protein quality, diversity and distributional similarity.
>
> We evaluate **protein quality** through sequence-based (ProGen 700M perplexity, ESM-2 650M pseudoperplexity, and BLAST) and structure-based metrics (ESMFold pLDDT, TM-score, and scPerplexity). These metrics complement each other because structure-based metrics may give false negatives in intrinsically disordered proteins, while sequence-based metrics, like perplexity can be tricked by nonsense repetitive sequences. Together, they provide a more complete picture of protein quality.
>
> *(continued in the next comment)*

---

> ### Author Response · Authors · 2024-11-27
>
> We evaluate the **diversity** by considering four aspects:
> - inner diversity ($CD_{0.5}$), or how diverse are batches of generated sequences?
> - duplicates ($CD_{0.95}$), or how many (near-)duplicates are in generated sequences?
> - novelty ($PCD_{0.5}$, $NCD_{0.5}$, Novelty), or how much do genereted sequences differ from the training data in terms of sequence identity?
> - repetitiveness ($Rep$), or what is the fraction of repetitive regions in the sequences?
>
> $CD_{0.5}$, $CD_{0.95}$, $PCD_{0.5}$, and $NCD_{0.5}$ are clustering-based approached, where the sequences are clustered according to their sequence identy % threshold. Novelty measures sequence identity between each sequence in a batch of generated sequences and the nearest neighbors in the training set.
>
>
> To measure the degree of resemblance between the distributions of training data and generated samples we use **distribution similarity** metrics using both sequence-based (ProtT5) and structure-based (ProteinMPNN) encoders. These metrics are Frechet distance (FD-seq, FD-struct), maximum mean discrepancy (MMD-seq, MMD-struct), and 1-Wasserstein optimal transport (OT-seq, OT-struct).
>
> This set of metrics covers the task of the evaluation from multiple perspectives. More details can be found in the **Section 4.1** and **Appendix B** of the revised manuscript.
>
> >**Q4**. Why the authors choose ESM2-8M - a very small representation learning model as the encoder across the study. As a usual practice, most of works use 650M model. Could the author reason the necessity of using 8M model? Moreover, as a control comparison, in Table 3, 10, 11, the authors combine “DiMA-8M” with other ESM2 encoders while in the main experiment, the authors claims using DiMA-33M. How do the authors explain the discrepancy between? Or is this a typo?
>
> Our work aims to **systematically explore architectural choices** in protein diffusion models. We deliberately used modest model sizes to enable comprehensive ablation studies and fair baseline comparisons within computational constraints. The 8M diffusion model experiments specifically probe encoder scaling effects, demonstrating that even a small diffusion model benefits from larger encoders up to 650M parameters - an important finding for future scaling efforts. For main experiments comparing with baselines, we use a 33M parameter model that balances computational demands with model capacity.
>
> >**W2**. Inconsistent results presentation. In Tables 4 and 5 of the main text, the bold digits intended to denote the best results (i guess) are inconsistently labeled. This lack of clarity can lead to confusion about which results are truly the best and detracts from the paper’s credibility.
>
> Thank you for noting the inconsistent results presentation. We acknowledge this issue and we have revised the tables to clearly indicate which results are best relative to dataset values versus absolute best values.
>
> >**W5**. The sections/paragraphs are badly organized and hard to follow, please consider improve the manuscript during rebuttal.
>
> We appreciate your suggestions for improving the manuscript organization. We have reorganized the manuscript for clarity by rearranging the experiments section into clear thematic blocks. We have added a roadmap paragraph at the start of the experiments section to guide readers through.
>
> ---
>
> We thank you for your thoughtful feedback and suggestions. We would be grateful if you would consider raising your score, in case we have addressed your concerns. Please let us know if any questions still need clarification.
>
> ---
>
> [1] Zhang, Sitao et al. PRO-LDM: Protein Sequence Generation with a Conditional Latent Diffusion Model. bioRxiv,2024.
>
> [2] Lu, Amy X. et al. Tokenized and Continuous Embedding Compressions of Protein Sequence and Structure. bioRxiv, 2024.
>
> [3] Durairaj, Janani et al. What is hidden in the darkness? Deep-learning assisted large-scale protein family curation uncovers novel protein families and folds. bioRxiv, 2023.
>
> [4] Alamdari, Sarah et al. Protein generation with evolutionary diffusion: sequence is all you need. bioRxiv, 2024.

---

> > ### Comment · Reviewer_GJwR · 2024-12-02
> >
> > Thanks for addressing my concerns and questions. I have increased my score to 5 and adjusted my evaluation correspondingly :) I hope the discussion help improve the quality of the manuscript, especially for the prior presentation issues.

---

### Official Review · Reviewer_sQ4r · 2024-11-03

**Soundness:** 3
**Presentation:** 3
**Contribution:** 3
**Rating:** 6
**Confidence:** 3

**Summary:**

In this manuscript, the authors introduce a latent diffusion method that leverages representations derived from the protein language model, ESM-2, to design proteins by generating amino acid sequences. Extensive ablation studies and evaluations based on diverse matrics were performed by the authors showing the method's capacity in generating protein sequences. This study shows the latent diffusion based method shows practical benefits for protein sequence design in terms of computational efficiency and sequence diversity.

**Strengths:**

The authors present comprehensive ablation studies and evaluations using multiple metrics to demonstrate their method's effectiveness in generating protein sequences. Their results show the method's ability to generate diverse sequences while maintaining structural fidelity to the protein space. The inclusion of distributional similarity metrics is interesting. This evaluation framework offers valuable insights for the protein sequence design community and establishes useful benchmarking standards.

**Weaknesses:**

The manuscript does not sufficiently justify why latent diffusion-based sequence design methods would provide advantages over alternative approaches, particularly discrete diffusion methods, in the context of protein sequence design. The authors may provide more theoretical or practical benefits of their latent diffusion approach. To strengthen the work, the author should provide additional evaluation metrics including sequence novelty, and provide more discussion about the underlying relationships between the metrics they used, clarify how they complement or potentially contradict each other.

**Questions:**

Several critical points require attention in the manuscript:

[Technical Motivation and Advantages]

While the manuscript identifies protein sequence design as an important problem, it lacks sufficient justification for using latent diffusion over established methods like discrete diffusion. The technical advantages and unique capabilities of latent diffusion in this specific context should be pointed out.


[Evaluation Metrics]
1. A novelty assessment comparing generated sequences with native sequences (e.g., sequence identity analysis) should be provided. One major concern is that the high-performance metrics could result from training set memorization rather than genuine generative capabilities. It is suggested to calculate sequence identity between generated sequences and their nearest neighbors in the training set.
2. The choice of Progen perplexity (PPL) as a quality metric requires justification. The authors should provide more discussion to clarify whether PPL is reliable for protein sequence design (ref: https://arxiv.org/abs/2210.05892).
3. Sequence length: The evaluation is limited to a narrow sequence length window (128-254 amino acids). This restriction raises important questions about method generalizability: How does performance scale with longer sequences? Could discrete diffusion methods demonstrate superior quality and diversity for longer sequences?

[Model Scaling Effects]

The observed decrease in distribution fitness scores (FD-seq, MMD-seq) when using the ESM3b encoder appears to contradict previous findings about model scaling benefits. This unexpected result warrants deeper analysis and discussion.

---

> ### Author Response · Authors · 2024-11-27
>
> We sincerely thank the reviewer for their thorough and constructive feedback. We appreciate the recognition of our comprehensive evaluation framework and its potential value for the protein design community. Below, we address each point raised.
>
> **[Technical Motivation and Advantages]**
>
> >The manuscript does not sufficiently justify why latent diffusion-based sequence design methods would provide advantages over alternative approaches, particularly discrete diffusion methods, in the context of protein sequence design. The authors may provide more theoretical or practical benefits of their latent diffusion approach.
>
> >While the manuscript identifies protein sequence design as an important problem, it lacks sufficient justification for using latent diffusion over established methods like discrete diffusion. The technical advantages and unique capabilities of latent diffusion in this specific context should be pointed out.
>
> We agree that the manuscript would benefit from a clearer articulation of the technical advantages of latent diffusion. Our approach offers several key benefits:
>
> 1. The continuous nature of the latent space **enables direct application of established score-based techniques** like classifier and classifier-free guidance without requiring discrete approximations. This creates opportunities for more controlled and directed protein generation.
>
> 2. Training in a continuous latent space is generally **more stable and efficient** compared to discrete spaces. Recent work on continuous vs discrete latent spaces (e.g., CHEAP) has demonstrated that continuous representations typically capture richer and more informative features.
>
> 3. As demonstrated in our ablation studies, the continuous latent space **allows for fine-grained optimization** of multiple aspects of the diffusion process - from noise scheduling to architectural choices. This flexibility helped us identify crucial components for effective protein generation.
>
> **[Evaluation Metrics]**
>
> >To strengthen the work, the author should provide additional evaluation metrics including sequence novelty
>
> >A novelty assessment comparing generated sequences with native sequences (e.g., sequence identity analysis) should be provided. One major concern is that the high-performance metrics could result from training set memorization rather than genuine generative capabilities. It is suggested to calculate sequence identity between generated sequences and their nearest neighbors in the training set.
>
> We thank the reviewer for raising this important point. While we had **included clustering-based novelty analysis** in our evaluation framework (detailed in **Appendix B.3**), we acknowledge this could have been more prominently discussed in the main text.
>
> Here we provide a direct sequence identity analysis. We calculate sequence distance between each generated sequence and its nearest neighbor in the training dataset and report the mean over 2048 pairs of sequences. The resulting novelty score is  $Novelty = 1 - Distance$, so Novelty = 0 means absolute memorization of the dataset. Our results (**Tables 1 and 2 below**) show that DiMA achieves a novelty score of 35.7% on SwissProt (compared to the dataset's internal novelty of 25.3%) and 72.9% on AFDBv4-90 (dataset: 57.7%). These scores indicate that our model generates novel sequences while maintaining meaningful similarity to natural proteins.
>
> We also perform co-clustering analysis (**Appendix B.3**) of generated sequences with dataset sequences using MMseqs2 at 50% sequence identity threshold. This analysis yields two metrics: $PCD_{0.5}$ and $NCD_{0.5}$, representing the ratios of "positive" clusters (containing both generated and dataset sequences) and "negative" clusters (containing only generated sequences) to the total number of sequences, respectively. The balance between these metrics ($PCD_{0.5} = 0.246$, $NCD_{0.5} = 0.392$ for SwissProt) demonstrates that our model both captures the distribution of known protein families and generates novel sequences that form their own clusters. Importantly, sequences in negative clusters maintain high structural plausibility (as measured by pLDDT, perplexity, scPerplexity and TM-scores), indicating successful generalization beyond the training data while preserving biological relevance.

---

> > ### Comment · Reviewer_sQ4r · 2024-11-27
> > **Response to the rebuttal**
> >
> > Many thanks to the authors for the clustering-based novelty analysis result. It's interesting as the generated sequences capture known protein family distributions and form novel clusters. My concerns are resolved with the provided info. I'll raise my rating.

---

> ### Author Response · Authors · 2024-11-27
>
> Table 1. Diversity evaluation of models trained on SwissProt dataset.
> | Model | Novelty (↓) | $CD_{0.5}$ (↑) | $CD_{0.95}$ (↑) | $PCD_{0.5}$ | $NCD_{0.5}$ |
> |--------|---------|-----------|------------|------------|---------|
> | Dataset (SwissProt)     | 74.7 | 0.000 | 0.943 | 0.990 | 0.304 |
> | Random sequences     | 14.9 | 0.000 | 1.000 | 1.000 | 0.000 |
> | RITA                 | 30.6 | 0.988 | 0.998 | 0.125 | 0.861 |
> | EvoDiff-OADM         | 22.4 | 0.986 | 1.000 | 0.058 | 0.929 |
> | DPLM                 | 88.4 | 0.494 | 0.812 | 0.267 | 0.236 |
> | nanoGPT             | 46.2 | 0.900 | 0.994 | 0.226 | 0.679 |
> | DiMA                 | 64.3 | 0.611 | 0.992 | 0.246 | 0.392 |
>
> Table 2. Diversity evaluation of models trained on AFDBv4-90 dataset.
> | Model | Novelty (↓) | CD₀.₅ (↑) | CD₀.₉₅ (↑) | PCD₀.₅ | NCD₀.₅ |
> |--------|---------|-----------|------------|------------|---------|
> | Dataset (AFDB)        | 42.4 | 0.994 | 1.0 | 0.029 | 0.966 |
> | Random sequences         | 15.3 | 1.0 | 1.0 | 0.0 | 1.0 |
> | nanoGPT                 | 30.8 | 1.0 | 0.986 | 0.037 | 0.953 |
> | DPLM                     | 48.4 | 0.97 | 0.476 | 0.132 | 0.341 |
> | DiMA                     | 27.1 | 1.0 | 1.0 | 0.0 | 0.992 |
>
> We have made these analyses more prominent in the revised manuscript and included the detailed results tables for both datasets.
>
> >The choice of Progen perplexity (PPL) as a quality metric requires justification. The authors should provide more discussion to clarify whether PPL is reliable for protein sequence design (ref: https://arxiv.org/abs/2210.05892).
>
> We appreciate the reviewer raising this important point about the reliability of perplexity (ppl) as an evaluation metric, and the reference to known limitations of ppl in text generation evaluation. We agree that no single metric is perfect for evaluating protein sequence quality, which is why **we employ a diverse suite of complementary metrics**. Let us elaborate on our rationale:
>
> A key weakness of perplexity is its tendency to assign low (better) values to low-information, repetitive sequences. However, such sequences would perform poorly on our structural metrics (pLDDT, TM-score, scPerplexity) since repetitive sequences typically do not fold into stable structures. Our Rep metric directly quantifies the degree of repetition on subsequences of different sizes. This helps identify cases where low perplexity might be artificially achieved through repetitive patterns.
>
> Conversely, structure-based metrics may underperform when evaluating intrinsically disordered proteins (IDPs) that lack stable 3D structures. Here, sequence-based metrics like perplexity provide complementary information. scPerplexity adds another layer of validation by measuring the consistency between sequence and structure predictions.
>
> We observe that our method performs well across all metrics simultaneously, suggesting genuine generation quality rather than exploitation of any single metric's weaknesses. For instance, while DPLM shows excellent perplexity scores (even better than the dataset), its high repetition rate (Rep = 0.781 vs. 0.045 for the dataset) suggests potential quality issues that are caught by our comprehensive evaluation framework.
>
> We have expanded the discussion of evaluation metrics in the revised manuscript, including a detailed analysis of their complementarity and potential limitations. This includes specific consideration of perplexity's strengths and weaknesses in the context of protein sequence evaluation.
>
> >provide more discussion about the underlying relationships between the metrics they used, clarify how they complement or potentially contradict each other.
>
> Thank you for suggesting a deeper discussion of metric relationships. In our evaluation framework, we use three categories of metrics that assess different aspects of protein generation:
>
> 1. Protein quality metrics:
>    - Structure-based:
>      * pLDDT: Measures confidence in predicted 3D structure positions
>      * TM-score: Evaluates structural similarity to known proteins
>      * scPerplexity: Assesses sequence-structure consistency
>    - Sequence-based:
>      * Progen perplexity: Evaluates sequence naturalness
>      * BLAST identity: Measures similarity to known sequences
>
> These metrics complement each other because structure-based metrics may miss quality issues in intrinsically disordered regions, while sequence-based metrics might not capture structural plausibility. Together, they provide a more complete picture of protein quality.
>
> 2. Distribution similarity metrics:
>    - Sequence representation-based:
>      * FD-seq: Measures overall distribution matching
>      * MMD-seq: Sensitive to mode coverage
>      * OT-seq: Evaluates optimal pairing costs
>    - Structure representation-based:
>      * FD-struct, MMD-struct, OT-struct: Analogous metrics using structural encodings
>
> These metrics provide a perspective on how well the generated distribution matches the training distribution.

---

> ### Author Response · Authors · 2024-11-27
>
> 3. Diversity metrics:
>    - Rep: Quantifies internal sequence repetition
>    - CD₀.₅, CD₀.₉₅: Measure sequence diversity at different identity thresholds capturing overall diversity and mount of (near-)duplicates
>    - PCD₀.₅, NCD₀.₅: Assess overlap with training distribution
>    - Novelty: Measures sequence identity with nearest neighbors in the dataset.
>
> These metrics can reveal different types of generation issues:
> - High Rep captures repetitions to which language models can be prone. In concert with low perplexity high Rep suggests artificial quality through repetition (as seen in DPLM)
> - Low CD₀.₉₅ indicates duplicate generation
> - Imbalanced PCD₀.₅/NCD₀.₅ suggests either overfitting or mode collapse
> - High sequence identity (through novelty score) indicates issues with memorization of the training data
>
> >Sequence length: The evaluation is limited to a narrow sequence length window (128-254 amino acids). This restriction raises important questions about method generalizability: How does performance scale with longer sequences? Could discrete diffusion methods demonstrate superior quality and diversity for longer sequences?
>
> We appreciate the reviewer raising this important question about generalizability to longer sequences. The current length restriction (128-254 amino acids) was chosen primarily for practical reasons. This length range covers many functionally important protein domains and families, yet enables thorough evaluation across multiple metrics, comparison with multiple baselines trained from scratch to match parameter count and extensive ablation studies within available computational resources.
>
> We agree that evaluating longer sequences would be valuable, and we believe our approach has good potential for scaling to longer sequences for several reasons. First, our model's transformer-based design is inherently capable of handling variable-length inputs. The primary limitation is computational rather than architectural. Second, our ablation studies of various components (noise scheduling, self-conditioning, etc.) provide insights that should transfer to longer sequence generation. And third, recent advances in efficient latent spaces (e.g., CHEAP) demonstrate the possibility of reducing sequence length dimensionality without significant quality degradation. This suggests a path toward handling longer sequences without proportional increase in computational costs.
>
> We commit to exploring longer sequence generation in future work, particularly leveraging emerging efficient latent space representations. For the current study, we prioritized comprehensive evaluation and ablation studies within a manageable length range to establish solid foundational understanding of the method's capabilities.
>
> **[Model Scaling Effects]**
>
> >The observed decrease in distribution fitness scores (FD-seq, MMD-seq) when using the ESM3b encoder appears to contradict previous findings about model scaling benefits. This unexpected result warrants deeper analysis and discussion.
>
> Thank you for highlighting this interesting observation about distributional scores with the ESM-2 3B encoder. While it might appear counterintuitive at first, we believe this result reveals important insights about the interplay between model capacity and latent space complexity.
>
> The observed behavior reflects a fundamental trade-off between quality and diversity in our current setup. With ESM-2 3B, we see improved quality metrics (pLDDT increased from 71.5 to 74.6, ppl improved from 9.53 to 8.52), but this comes with decreased diversity ($CD_{0.5}$ drops from 0.748 to 0.660). This suggests our 8M parameter diffusion model we use for these scaling experiments is reaching a capacity limit where improvements in one aspect come at the cost of another.
>
> This trade-off emerges because ESM3b produces richer, more complex representations that our 8M parameter diffusion model must compress. Given limited capacity, the model optimizes for generation quality in a smaller region of the latent space (leading to better quality metrics) rather than attempting broader but lower-fidelity coverage (yielding better distribution metrics).
>
> These findings have important implications for model scaling. They suggest that to fully leverage larger encoders like ESM-2 3B, we likely need to scale up the diffusion model accordingly. The current results represent a capacity-constrained optimization where the model must balance quality and coverage. Future work with larger diffusion models might achieve improvements in both aspects simultaneously.
>
> We expand this analysis in the revised manuscript to better explain these relationships and their implications for model scaling (**Section 4.3**).
>
> ---
>
> We appreciate the thoughtful suggestions for improving the manuscript. We would be grateful if you would consider raising your score, in case we have addressed your concerns. Please let us know if any questions still need clarification.

---

### Official Review · Reviewer_w8qc · 2024-11-03

**Soundness:** 3
**Presentation:** 3
**Contribution:** 3
**Rating:** 6
**Confidence:** 4

**Summary:**

This work introduces DiMA, a protein sequence generation model by performing latent diffusion on ESM2-650M embeddings. The model is evaluated by sequence diversity, BLAST identity to known sequences, perplexity under autoregressive models, and distribution matching to embeddings of protein families. Structure sensibility is also assessed by examining the pLDDT and self-consistency perplexity. Some emphasis is also given to methodological design, including ablating the noise schedule used, self-conditioning, ESM2 encoding, time embedding information insertion into the model. When compared to several baselines that include autoregressive, energy-based, and diffusion-based sequence generation methods, trained to have the same parameter count and trained on the same datasets,

**Strengths:**

* Ablation results are thorough and provide insight for others venturing into generating proteins via their latent embeddings.
* Results are comprehensive. Retraining baselines on the same dataset & controlling for parameter size is a good and rigorous effort.
* On the whole, sequence generation results are quite strong against baselines.
* Focusing on distribution similarity is IMO a great standard for this community, and it's nice to see that many of these metrics perform well.

**Weaknesses:**

* The encoder compression sections feels underdeveloped. At minimum, it should probably include a citation to CHEAP (https://www.biorxiv.org/content/10.1101/2024.08.06.606920v1.full.pdf). A stretch experiment would be to see how sequence diffusion performance changes when doing so in the CHEAP embedding space. ESM 3B yields better results in Table 3, but we know that larger embedding dimensions is generally harder for latent diffusion models, so all of this feels a bit counter intuitive. I think at this point it's not too much of a mystery that we can do sequence generation by diffusion on ESM embeddings, and ample literature exists on diffusion design choices, but the finer details of how information is captured in these latent embeddings is more under-explored, and including it in this paper would strengthen its utility.
* "Most comprehensive evaluation" in the introduction is a strong claim. The evaluations are indeed very impressive, but adding a qualifier like this is rather misleading, and does not promote the paper's longevity.
* I think the work has a lot of results, which is great, but they feel a bit under-analyzed. Reducing comparisons to single valued points without any examples of generated sequences (or their folded structures) feels short of convincing.  In future iterations, the work could be strengthened by more biological and qualitative analyses like those on pages 24-28, and moving that into the main text. In its current form, the captions are vague; for e.g. Figure 11, 12, 13 especially.
* I think it's productive to have figures like Appendix B.2 which demonstrate distribution comparisons in the main text. Though table space is limited, it would be nice to report the stds from the mean if possible.
* In the main baseline tables, is it possible to also add the length distributions of baselines and generated sequences, given how important they are for many of the reported metrics?

Minor:
* Line 127: "High-generalizing" reads slightly awkwardly
* This is not critical for me to raise my score, but could I think the results section would be better organized with better subsections, such that a study is presented right before the results. Found it a bit hard to jump between the sections and find the correct \paragraph{} heading.
* fontsize inconsistencies in tables could be polished
* noise schedule formula is duplicated between main text and Appendix D.1

**Questions:**

* In the data section, could authors clarify how they'd used the AFDB dataset, since it is a structural dataset? Why not just train on UniRef or another similarly sized whole-protein dataset?

Minor:
* line 237: description of "ProteinMPNN structure representations" is a bit murky. Which layer was used? Which ProteinMPNN structure? Where did the structure come from given that the generation is for sequences?

---

> ### Comment · Reviewer_w8qc · 2024-11-26
>
> I'm also happy to further participate in the discussion period, if the authors would like to post replies to the reviews.

---

> ### Author Response · Authors · 2024-11-27
>
> Thank you for your thoughtful and detailed feedback. We deeply appreciate the time you have taken to review our work and your positive comments about our ablation studies, comprehensive results, and rigorous baseline comparisons. We aim to address your concerns and questions below.
>
> >**W1 (part1)**. The encoder compression sections feels underdeveloped. At minimum, it should probably include a citation to CHEAP (https://www.biorxiv.org/content/10.1101/2024.08.06.606920v1.full.pdf). A stretch experiment would be to see how sequence diffusion performance changes when doing so in the CHEAP embedding space.
>
> We are particularly grateful for your insightful suggestion regarding the encoder compression section. Following your recommendation, we have conducted additional experiments using CHEAP encoders [1]. We put to test two variants: CHEAP_shorten_1_dim_64 and CHEAP_shorten_2_dim_64. Both encoders compress one dimension to 64, but CHEAP_shorten_2_dim_64 additionally reduces the sequence length dimension by half. For these experiments, we only replaced ESM-2 with CHEAP encoders while keeping all other aspects of our architecture and training procedure exactly the same, using the optimal settings discovered through our ablation studies. The results are remarkably strong:
>
> Table 1. Performance comparison between DiMA with ESM-2 and CHEAP encoders and selected baselines. FD-seq measures distributional similarity with the dataset, pLDDT and ppl show protein quality, Rep, CD₀.₅ and Novelty evaluate diversity.
> | Encoder | FD-seq (↓) | pLDDT (↑) | Progen ppl (↓) | Rep (↓) | CD₀.₅ (↑) | Novelty (↑) |
> |--------------|------------|-------------|------------|----------------|----------|----------|
> | DiMA CHEAP_shortcn_1_dim_64 | 0.32 | 80.3 | 7.05 | 0.053 | 0.572 | 49.4 |
> | DiMA CHEAP_shortcn_2_dim_64 | 0.36 | 81.4 | 7.14 | 0.041 | 0.561 | 50.7 |
> | DiMA ESM-2| 0.34 | 83.3 | 5.07 | 0.320 | 0.611 |  35.7 |
> | DPLM | 0.50 | 84.1 | 3.57 | 0.781 | 0.494 | 11.6 |
> | nanoGPT | 1.24| 61.0 | 8.87 | 0.228 | 0.900 |  53.8 |
>
> Both CHEAP variants achieve impressive performance - their pLDDT scores (80.3 and 81.4) closely match the dataset quality (80.7), and their FD-seq metrics (0.32 and 0.36) are comparable with DiMA (0.34) while significantly outperforming other baselines. The fact that we achieved these results without any modifications to our architecture or training procedures validates our extensive ablation studies and demonstrates that our insights about latent diffusion for protein generation generalize well across different embedding spaces. This opens up exciting possibilities for the community to develop new protein design models based on continuous latent diffusion. Moreover, we believe even better results could be achieved by specifically tuning hyperparameters for CHEAP encoders.
>
> We are very grateful to you for suggesting this valuable research direction.
>
> >**W1 (part2)**. ESM 3B yields better results in Table 3, but we know that larger embedding dimensions is generally harder for latent diffusion models, so all of this feels a bit counter intuitive. I think at this point it's not too much of a mystery that we can do sequence generation by diffusion on ESM embeddings, and ample literature exists on diffusion design choices, but the finer details of how information is captured in these latent embeddings is more under-explored, and including it in this paper would strengthen its utility.
>
> Thank you for highlighting this important observation. We agree that the relationship between embedding dimension and model performance deserves deeper analysis. Our results with ESM 3B show an interesting quality-diversity tradeoff that may explain the counter-intuitive scaling behavior. While quality metrics improve (pLDDT increases from 71.5 to 74.6, perplexity improves from 9.53 to 8.52), we observe decreased diversity ($CD_{0.5}$ drops from 0.748 to 0.660, FD-seq increases from 0.266 to 0.279).
>
> We believe this reflects our 8M parameter diffusion model reaching a capacity bottleneck when handling the richer, higher-dimensional ESM 3B representations. Given limited capacity, the model appears to optimize for higher quality generation in a more restricted region of the latent space rather than maintaining broader coverage with lower fidelity. This suggests that fully leveraging larger encoders like ESM 3B likely requires correspondingly scaling up the diffusion model to maintain both quality and diversity simultaneously.
>
> We expand this analysis in the manuscript (Section 4.3) to explore these capacity-representation interactions, as they provide important insights into how information is captured and utilized across different scales of latent embeddings. This could help guide future work on model scaling strategies for protein generation.

---

> ### Author Response · Authors · 2024-11-27
>
> >**W2**. "Most comprehensive evaluation" in the introduction is a strong claim. The evaluations are indeed very impressive, but adding a qualifier like this is rather misleading, and does not promote the paper's longevity.
>
> We fully agree with this feedback. We have revised this to simply describe the specific evaluations we performed across sequence quality, diversity, novelty, distribution matching, and biological relevance.
>
> >**W3**. I think the work has a lot of results, which is great, but they feel a bit under-analyzed. Reducing comparisons to single valued points without any examples of generated sequences (or their folded structures) feels short of convincing. In future iterations, the work could be strengthened by more biological and qualitative analyses like those on pages 24-28, and moving that into the main text. In its current form, the captions are vague; for e.g. Figure 11, 12, 13 especially.
>
> We completely agree that the paper would benefit from more detailed qualitative analysis in the main text. While we provide all generated sequences in our repository, incorporating examples and their analysis directly in the paper would make our results more tangible and convincing.
>
> We have revised the figure captions to provide clear, detailed explanations of what each visualization shows. Thank you for bringing this to our attention.
>
> >**W4**. I think it's productive to have figures like Appendix B.2 which demonstrate distribution comparisons in the main text. Though table space is limited, it would be nice to report the stds from the mean if possible.
>
> Thank you for this suggestion. While space constraints prevent moving the original figure to the main text, we have expanded this analysis by adding Figure 4 in **Appendix B.3**, which comprehensively visualizes the distribution of distances between optimal pairs for all models and ablations. We also expand the corresponding discussion.
>
> >**W5**. In the main baseline tables, is it possible to also add the length distributions of baselines and generated sequences, given how important they are for many of the reported metrics?
>
> Thank you for this suggestion. Most models in our comparison, including DiMA, require sequence length to be specified prior to generation. For fair comparison, we sample lengths from the dataset's distribution for all models and random sequences. The only exceptions are models that generate autoregressively, but even for these models, we observe that the sequence lengths closely follow the dataset distribution. Therefore, adding length distributions to the tables would be redundant as they match the dataset distribution by design. We have made this sampling procedure more explicit in the methods section.
>
>
> >**W_minor**. Line 127: "High-generalizing" reads slightly awkwardly; This is not critical for me to raise my score, but could I think the results section would be better organized with better subsections, such that a study is presented right before the results. Found it a bit hard to jump between the sections and find the correct \paragraph{} heading.; fontsize inconsistencies in tables could be polished; noise schedule formula is duplicated between main text and Appendix D.1
>
> Thank you for bringing this to our attention. We have revised the manuscript in accordance with your recommendations. Escpecially, we have reorganized the manuscript for clarity by rearranging the experiments section into clear thematic blocks.
>
> >**Q1**. In the data section, could authors clarify how they'd used the AFDB dataset, since it is a structural dataset? Why not just train on UniRef or another similarly sized whole-protein dataset?
>
> We use AFDBv4_90 [3] which is a subset of AFDB (AlphaFold DB). It includes the UniRef50 sequences with AlphaFold-predicted pLDDT > 90. This filtering ensures that any quality issues in generated sequences stem from model architecture or training/sampling procedure, not training data quality.
>
> >**Q2**. line 237: description of "ProteinMPNN structure representations" is a bit murky. Which layer was used? Which ProteinMPNN structure? Where did the structure come from given that the generation is for sequences?
>
> Thank you for pointing this out. When we calculate distributional metrics the features are obtained either by embedding the sequences using the ProtT5 or the ESMFold-predicted structures using ProteinMPNN. We use the features from the last (3rd) layer from the backbone decoder with global average pooling applied along the length dimension to ensure a fixed-size embedding. The resulting vector from ProteinMPNN has size 128.
>
> ---
>
> We thank you for your thoughtful feedback and suggestions. We would be grateful if you would consider raising your score, in case we have addressed your concerns. Please let us know if any aspects still need clarification.
>
> ---
>
> [1] https://www.biorxiv.org/content/10.1101/2024.08.06.606920v1
>
> [2] https://www.biorxiv.org/content/10.1101/2023.03.14.532539v3

---

> > ### Author Response · Authors · 2024-12-03
> >
> > Dear Reviewer w8qc,
> >
> > Thank you again for your time and suggestions you have provided. Considering the approaching end of the rebuttal period, we would like to get your feedback and answer any further questions if any.
> >
> > Best regards,
> > Authors

---

### Official Review · Reviewer_bZ1a · 2024-11-04

**Soundness:** 3
**Presentation:** 2
**Contribution:** 3
**Rating:** 5
**Confidence:** 4

**Summary:**

This work presents an approach for unconditional protein sequence generation that operates via continuous diffusion in the latent space of protein sequence embeddings derived from the encoding model ESM2. Generated latent vectors are decoded back to amino acid sequence space via a lightweight decoder. By leveraging a pre-trained encoder like ESM2, the size of the backbone network driving the latent space diffusion process is relatively small. The authors present a series of evaluations that attempt to quantify the distribution modeling capabilities of their approach and the quality of sampled generations, including relative to internal ablations and other approaches trained on the SwissProt dataset (evaluations on AFDB are not presented relative to other approaches).

Post rebuttal: The authors' rebuttal addresses my primary concerns, and the promised revision promises to correct the significant instances of imprecision in the manuscript. I will adjust my scores accordingly.

**Strengths:**

While many works have presented compelling approaches for unconditional protein sequence generation, including sequence-based generative models (ProGen, ProtGPT, EvoDiff, etc), and latent diffusion models (eg ProLDM), to my knowledge this is a work that leverages the pre-trained ESM to define a latent space over which to learn the diffusion process. This is a valuable idea to contribute to the field, but it must hold in terms of evaluations. It is also interesting to provide a head-to-head comparison of discrete diffusion over amino acid sequences versus continuous diffusion over the latent sequence embeddings provided by ESM.

**Weaknesses:**

There are several weaknesses to this work:
- The claim that “the foundational task of unconditional generation remains underexplored and underappreciated” is simply incorrect — many works (e.g., ProGen, ProtGPT, EvoDiff) have explored and evaluated performance of sequence-based unconditional generation.
- Discussion of related works on latent diffusion for proteins, such as Pro-LDM and CHEAP, is missing from the Introduction and Related Works. In general, many related works on diffusion models for protein generation are not discussed.
- The claim that “The straightforward adaptation of image-based diffusion models has not yet yielded satisfactory results in this domain,” is also unfounded. In fact, diffusion-based models for protein generation have been proven state-of-the-art in structure space; in sequence space, discrete diffusion is not at all a straightforward adaptation of image-based diffusion. This is not directly applying existing image-based diffusion, as the authors imply. This imprecision signals an underlying weakness of the work and the authors’ claims.
- The claim “most comprehensive set of metrics” is a gross overclaim and significantly weakens the credibility of the paper.
- All the distributional metrics presented seem to be computed relative to the train set — this calls into question the strength and credibility of the evaluations, since such comparisons to train will only reflect the ability of the generative models to memorize the train set. Such distributional evaluations should be computed relative to an independent test set.
- The presence of repeated amino acid subsequences is not a metric of “diversity”; similarly, the 95% sequence identity threshold as the barrier to novelty is extremely lenient.
- The claims that DiMA performs as well or better than the benchmarked approaches is not upheld by the evaluations, where DiMA is often outperformed.
- The training and comparisons using SwissProt are not compelling, given the small (~200-300K sequences) size of SwissProt, making it easy to memorize, especially considering the use of the training set in the distributional metrics.

**Questions:**

1. Why were the distributional metrics computed relative to the train set? How do the authors justify the use of these metrics as an appropriate means to benchmark the learning capacity and generalizability of the tested models?
2. What was the sequence-level diversity/novelty achieved, relative to natural sequences? The repeated subsequence metric does not reflect this, and the 95% identity threshold is extremely lenient.
3. What was the reasoning behind the use of the SwissProt dataset when it is relatively so small, almost certainly represented within the ESM pretraining set, and given its size very easy to memorize?
4. What are the relative strengths, limitations, and novelties of DiMA versus another latent diffusion approach, Pro-LDM? These are not discussed.
5. The evaluations of “biological relevance” do not demonstrate evidence of real-world utility of the approach. It is not surprising that high homology is achieved given training on SwissProt. Could the authors comment on the ability of DiMA to generate new sequences relevant to an independent downstream evaluation or task?

---

> ### Author Response · Authors · 2024-11-21
>
> Thank you for your review and comments. We appreciate your time and effort and would like to address the concerns and clarify some apparent misunderstandings with following responses.
>
> > **W5**. All the distributional metrics presented seem to be computed relative to the train set — this calls into question the strength and credibility of the evaluations, since such comparisons to train will only reflect the ability of the generative models to memorize the train set. Such distributional evaluations should be computed relative to an independent test set.
>
> >**Q1**. Why were the distributional metrics computed relative to the train set? How do the authors justify the use of these metrics as an appropriate means to benchmark the learning capacity and generalizability of the tested models?
>
> Thank you for raising this important concern about the evaluation methodology. You are absolutely right that the use of proper test sets should be explicitly stated in the paper - this was an oversight in our presentation. To clarify: throughtout the work **we use a carefully constructed hold-out test set**, created using standard sequence identity filtering ($50\\%$ identity, $80\\%$ overlap in pairwise alignments using MMseqs2, **Appendix A**). Importantly, our test set shares the same underlying distribution as the training set, as evidenced by the distributional metrics between samples from both sets (see **Tables 2 and 8**, where "Dataset" entries show minimal divergence). This ensures that we evaluate both the quality of generated sequences and the model's ability to capture the true protein sequence distribution, rather than comparing against external datasets with potentially different distributions.
> We will rewrite the section of experiment description to state explicitly, that we use a test set for evaluation of the distributional metrics.
>
> >**W6**. The presence of repeated amino acid subsequences is not a metric of “diversity”; similarly, the 95% sequence identity threshold as the barrier to novelty is extremely lenient.
>
> >**Q2**. What was the sequence-level diversity/novelty achieved, relative to natural sequences? The repeated subsequence metric does not reflect this, and the 95% identity threshold is extremely lenient.
>
> We appreciate your concerns regarding the diversity metrics and the 95% sequence identity threshold. You raise a valid point that warrants better explanation in our manuscript.
>
> Our approach to diversity evaluation assesses the model's capacity to generate distinct protein variants while avoiding redundant outputs.  We employ the **clustering density** metric ($CD_t$) at two sequence identity thresholds: $t=50\\%$ and $t=95\\%$ (described in **Appendix B.3**). $CD_t$ represents the ratio of sequence clusters at threshold t to the total number of generated proteins. $CD_{0.5}$ is an established metric for assessing broad sequence diversity [1], analogous to the widely-adopted TM-score threshold of 0.5 used in structure generation [2].
> While clustering at a moderate threshold ($50\\%$) reveals the model's ability to generate diverse proteins, individual clusters may still contain nearly identical sequences—an undesirable characteristic for generative models. Therefore, we complement our analysis with $CD_{0.95}$, which specifically identifies near-duplicate sequences. This dual-threshold approach provides a more comprehensive assessment of sequence diversity compared to single-metric evaluations.
>
> We also perform **co-clustering analysis** of generated sequences with the dataset sequences (described in Appendix B.3). This analysis yields two metrics: $PCD_{0.5}$ and $NCD_{0.5}$, representing the ratios of "positive" clusters (containing both generated and dataset sequences) and "negative" clusters (containing only generated sequences) to the total number of sequences, respectively. As shown in **Tables 9, 12, 15, and 16**, these metrics reveal that our model not only captures the distribution of known protein families (through $PCD_{0.5}$) but also generates novel sequences that form their own clusters (through $NCD_{0.5}$). Notably, the sequences in negative clusters maintain high structural plausibility as measured by pLDDT, ppl and scPerplexity scores, suggesting that our model can generalize beyond the training data while preserving biological relevance.
>
> To directly address concerns about potential **memorization**, we measure novelty by calculating the mean sequence distance between each generated sequence and its nearest neighbor in the training dataset.  Results in the following Table 1 and 2 show that  **DiMA achieves a strong balance between similarity to natural proteins and generation of novel sequences**. On SwissProt, DiMA's novelty score of $35.7\\%$ approaches the dataset's characteristics ($25.3\\%$), while maintaining high clustering diversity ($CD_{0.95} = 0.992$), indicating that our model effectively learns the natural protein sequence distribution without overfitting.

---

> ### Author Response · Authors · 2024-11-21
>
> Table 1. Diversity evaluation of models trained on **SwissProt** dataset.
> | Model | Novelty (↑) | $CD_{0.5}$ (↑) | $CD_{0.95}$ (↑) | $PCD_{0.5}$ | $NCD_{0.5}$ |
> |--------|---------|-----------|------------|------------|---------|
> | Dataset (SwissProt) | 25.3 | 0.000 | 0.943 | 0.990 | 0.304 |
> | Random sequences | 85.1 | 0.000 | 1.000 | 1.000 | 0.000 |
> | RITA | 69.4 | 0.988 | 0.998 | 0.125 | 0.861 |
> | EvoDiff-OADM | 77.6 | 0.986 | 1.000 | 0.058 | 0.929 |
> | DPLM | 11.6 | 0.494 | 0.812 | 0.267 | 0.236 |
> | nanoGPT | 53.8 | 0.900 | 0.994 | 0.226 | 0.679 |
> | DiMA | 35.7 | 0.611 | 0.992 | 0.246 | 0.392 |
>
> Table 2. Diversity evaluation of models trained on **AFDBv4-90** dataset.
> | Model | Novelty (↑) | $CD_{0.5}$ (↑) | $CD_{0.95}$ (↑) | $PCD_{0.5}$ | $NCD_{0.5}$ |
> |--------|---------|-----------|------------|------------|---------|
> | Dataset (AFDB) | 57.6 | 0.994 | 1.0 | 0.029 | 0.966 |
> | Random sequences | 84.7 | 1.0 | 1.0 | 0.0 | 1.0 |
> | nanoGPT | 69.2 | 1.0 | 0.986 | 0.037 | 0.953 |
> | DPLM | 51.6 | 0.97 | 0.476 | 0.132 | 0.341 |
> | DiMA  | 72.9 | 1.0 | 1.0 | 0.0 | 0.992 |
>
> ---
> >**W8**. The training and comparisons using SwissProt are not compelling, given the small (~200-300K sequences) size of SwissProt, making it easy to memorize, especially considering the use of the training set in the distributional metrics.
>
> > **Q3**. What was the reasoning behind the use of the SwissProt dataset when it is relatively so small, almost certainly represented within the ESM pretraining set, and given its size very easy to memorize?
>
> We appreciate your concern about SwissProt's size and potential limitations. You raise an important point that deserves careful explanation of our dataset selection rationale.
>
> As detailed in **Appendix A**, our work utilizes two datasets: SwissProt with 470k sequences and AFDBv4-90 with 2.2M sequences. The half-million sequences in SwissProt provide sufficient data for training, particularly when compared to other successful approaches in the field. For context, structure-based generative models like FoldFlow achieve strong results with just 22,248 proteins [3], and text generation models like Discrete Flow Models (DFMs) [4] use text8 dataset of similar size (100M tokens) to SwissProt (110M tokens) while requiring almost three times more parameters than our model. Moreover, SwissProt's manual curation ensures high-quality training data, making it an ideal choice for validating our approach. The strong performance on both SwissProt and the much larger AFDBv4-90 dataset demonstrates our model's ability to scale effectively while maintaining generation quality.
>
> Importantly, the novelty evaluation presented in Tables 1 and 2 above demonstrates that our **model does not simply memorize the training data** - it generates novel sequences while maintaining biologically relevant patterns present in natural proteins.
>
> ---
> >**W2**. Discussion of related works on latent diffusion for proteins, such as Pro-LDM and CHEAP, is missing from the Introduction and Related Works. In general, many related works on diffusion models for protein generation are not discussed.
>
> >**Q4**. What are the relative strengths, limitations, and novelties of DiMA versus another latent diffusion approach, Pro-LDM? These are not discussed.
>
> Thank you for bringing this important point to our attention. We agree that a discussion of PRO-LDM and other latent diffusion approaches was missing, and we will add them to the Introduction and Related Work sections.
>
> Our work differs from PRO-LDM in several aspects that we should have articulated more clearly:
> 1. While PRO-LDM jointly trains an autoencoder with its diffusion model, DiMA uses pre-trained protein language model encodings. This design choice both simplifies training and avoids potential issues with embedding collapse during joint training [5]. We provide empirical evidence showing that using context-dependent encodings rather than embeddings leads to superior results (Table 2).
> 2. Through comprehensive ablation studies (pages 5-6, Table 2, Appendices C.1 and C.2 ), we identify several critical components for effective protein latent diffusion:
> - We demonstrate that diffusion on encodings requires more aggressive noise scheduling than typical implementations
> - We show that self-conditioning substantially improves generation quality
> - We analyze the impact of various architectural choices
> These technical insights, while fundamental to model performance, were not investigated in previous work on protein latent diffusion.
> 3. While PRO-LDM focuses on family-specific generation tasks with relatively small datasets (up to 69k sequences), our work aims to develop a foundational model for protein sequence generation that can further be adapted to various downstream tasks. That is why we focus on tuning available degrees of freedom in diffusion model design and comparing against multiple baselines. Nonetheless, we  demonstrate family-specific generation through finetuning (Table 6).

---

> ### Author Response · Authors · 2024-11-21
>
> >**W3**. The claim that “The straightforward adaptation of image-based diffusion models has not yet yielded satisfactory results in this domain,” is also unfounded. In fact, diffusion-based models for protein generation have been proven state-of-the-art in structure space; in sequence space, discrete diffusion is not at all a straightforward adaptation of image-based diffusion. This is not directly applying existing image-based diffusion, as the authors imply. This imprecision signals an underlying weakness of the work and the authors’ claims.
>
> Thank you for bringing up this important point about our statement regarding image-based diffusion models. We acknowledge that our phrasing was imprecise and could lead to misunderstanding. To clarify: when we referred to "image-based diffusion models," we specifically meant the direct application of Gaussian diffusion on continuous representations, as pioneered by Ho et al. [6] in the image domain. You are absolutely right that discrete diffusion models for protein sequences represent a distinct approach and should not be characterized as adaptations of image-based methods.
>
> Our intent was to highlight that applying standard Gaussian diffusion approaches from the image domain to protein embeddings requires careful consideration and modification of various components (noise schedule, architecture, etc.) to achieve high-quality results. This is why we conducted extensive ablation studies to identify the optimal hyperparameters and architectural choices for the protein domain. We have revised this statement in the paper to more precisely convey our meaning and avoid any confusion with discrete diffusion approaches, which have indeed shown impressive results in protein structure generation.
>
> >**W1**. The claim that “the foundational task of unconditional generation remains underexplored and underappreciated” is simply incorrect — many works (e.g., ProGen, ProtGPT, EvoDiff) have explored and evaluated performance of sequence-based unconditional generation.
>
> Thank you for pointing out the imprecision in our statement about unconditional generation. We agree that numerous important works (including ProGen, ProtGPT, and EvoDiff) have made valuable contributions to unconditional protein sequence generation. We should have been more precise in articulating our meaning: while many approaches have been proposed for unconditional protein sequence generation, there is still room for progress in the field. Our experimental results, presented in Tables 4-5 and throughout the paper, support this harmless idea.
>
> We have revised this statement in the paper to more accurately reflect the current state of the field and better explain the motivation for our work.
>
> >**W4**. The claim “most comprehensive set of metrics” is a gross overclaim and significantly weakens the credibility of the paper.
>
> You are right about our claim regarding the "most comprehensive set of metrics." We appreciate you pointing this out. We will modify this statement to simply describe our evaluation framework, which includes structural quality metrics (pLDDT, TM-score, scPerplexity), sequence properties (perplexity, pseudoperplexity, BLAST identity), distributional similarity (FD, MMD, OT), and diversity measures ($CD_{0.5}$, $CD_{0.95}$, PCD, NCD, novelty).
>
> >**W7**. The claims that DiMA performs as well or better than the benchmarked approaches is not upheld by the evaluations, where DiMA is often outperformed.
>
> Thank you for this observation about our performance claims. We should be more precise in discussing DiMA's performance relative to other approaches. When evaluating generative models, it's crucial to note that the optimal value for many metrics is not necessarily the maximum or minimum, but rather the value that matches the training distribution. For example, on AFDB dataset, while DPLM achieves better (lower) perplexity score ($4.73$) than DiMA ($11.57$), the dataset's own perplexity is $10.83$. Distributional metrics and evaluation of the diversity suggests, that DPLM experiences mode collapse and repeats sequences of high quality ($CD_{0.95}=0.476$ vs $1.0$ for the dataset) and DiMA better reproduces dataset's distribution (FD-seq $0.27$ vs. $0.11$ for the dataset and $1.46$ for DPLM). This illustrates the quality-diversity trade-off, recently shown in [7] for image generation.

---

> ### Author Response · Authors · 2024-11-27
>
> >**Q5**. The evaluations of “biological relevance” do not demonstrate evidence of real-world utility of the approach. It is not surprising that high homology is achieved given training on SwissProt. Could the authors comment on the ability of DiMA to generate new sequences relevant to an independent downstream evaluation or task?
>
> Thank you for raising this point. We have substantially updated the biological relevance section in the main text and **Appendix D** to better articulate our evaluation approach and findings.
>
> Our analysis demonstrates that **DiMA effectively captures the functional diversity** of natural proteins in multiple ways. Using InterProScan annotation, we show that 92% of DiMA-generated sequences are annotated as known SUPERFAMILY members, closely matching the dataset's 80% annotation rate. Importantly, DiMA accurately reproduces the natural distribution of domain lengths, with most domains spanning 50-75 amino acids, and effectively models both structured regions and intrinsically disordered regions (IDRs), capturing the full spectrum of protein functionality. DPLM, on the other hand, tends to generate longer domains (around 250 residues).
>
> To directly address the concern about **downstream utility and novelty**, we conduct new proof-of-concept experiments on sequence **inpainting** - a conditional generation task that requires generating novel sequences that maintain structural and functional coherence. We evaluate DiMA on 180 sequences from our SwissProt test set, specifically selecting sequences with at most 50% identity to their nearest neighbors in the training set to prevent memorization effects. For each sequence, we mask random regions of variable length (8-50 amino acids) and assess generation quality using multiple stringent criteria: complete sequence pLDDT ≥ 80, inpainted region pLDDT ≥ 80, and preservation of unmasked region structure (RMSD ≤ 1Å compared to reference). We perform 10 generation attempts per sequence and consider generation successful if any attempt satisfied all quality criteria.
>
> DiMA achieves a 42.2% success rate on this task, outperforming both DPLM (40.0%) and random baseline (21.1%). The generated sequences show high novelty (inpainted region average novelty of 0.80) while maintaining structural plausibility (average pLDDT 66.9 vs 59.3 for DPLM), demonstrating that DiMA generates novel, functionally relevant sequences rather than simply memorizing training data. These results indicate that while DiMA was trained on SwissProt, it learns generalizable principles of protein sequence-structure relationships that enable generation of novel, biologically relevant sequences for downstream applications. We believe these findings provide strong evidence for the real-world utility of our approach.
>
> ---
>
> In conclusion, we appreciate your thorough review which has helped us identify the areas where our presentation needs improvement. As our responses demonstrate, many of your concerns stem from imprecise writing and insufficient explanation rather than fundamental flaws in the methodology or results. Our work introduces an approach to protein generation that achieves strong results, as supported by our experimental results. Our extensive ablation studies provide insights into the impact of various architectural choices and training strategies, that we believe will benefit the broader research community working on protein generation.
>
> We believe these clarifications address your main concerns and demonstrate that our work makes a valuable contribution to the field. Given that the core technical contribution and results remain sound, and that most issues can be addressed through improved presentation and additional context, we respectfully suggest that a "strong reject" may be too severe. We would be grateful if you would consider revising your assessment.
>
> ---
>
> [1] Suzek B., et al., UniRef: comprehensive and non-redundant UniProt reference clusters, Bioinformatics, 2007
>
> [2] Yim, Jason, et al., SE(3) diffusion model with application to protein backbone generation. arXiv, 2023.
>
> [3] Bose, Avishek, et al., SE(3)-Stochastic Flow Matching for Protein Backbone Generation. arXiv, 2024.
>
> [4] Campbell, Andrew, et al., Generative Flows on Discrete State-Spaces: Enabling Multimodal Flows with Applications to Protein Co-Design. arXiv, 2024.
>
> [5] Dieleman, Sander, et al., Continuous diffusion for categorical data. arXiv, 2022.
>
> [6] Ho, Jonathan, et al., Denoising Diffusion Probabilistic Models. arXiv, 2020.
>
> [7] Astolfi, Pietro, et al., Consistency-diversity-realism Pareto fronts of conditional image generative models. arXiv, 2024.

---

> ### Comment · Reviewer_bZ1a · 2024-12-02
> **Response to rebuttal**
>
> Thanks to the authors for their efforts in the rebuttal. The rebuttal addresses my primary concerns, and the promised revision promises to correct the significant instances of imprecision in the manuscript. I have increased my scores accordingly.

---

### Author Response · Authors · 2024-11-27
**Summary of Rebuttal**

Dear Reviewers, ACs and SACs,

We sincerely thank all reviewers for their efforts and constructive feedback. During the rebuttal period, we have addressed the raised concerns and significantly strengthened our work through several key improvements:

- Demonstrated DiMA's effectiveness with CHEAP encoders (pLDDT: 81.7, FD-seq: 0.309), showing strong performance without any architectural modifications (suggested by Reviewers bZ1a and w8qc)
- Added comprehensive comparisons with Chroma and MultiFlow as additional baselines (suggested by Reviewer fqCm)
- Conducted proof-of-concept experiments on sequence inpainting, achieving 42.2\\% success rate under stringent criteria (complete sequence pLDDT ≥ 80, inpainted region pLDDT ≥ 80, unmasked RMSD ≤ 1Å) (suggested by Reviewer PQDj)
- Validated structural quality metrics using both ESMFold and OmegaFold, confirming evaluation robustness (suggested by Reviewer PQDj)
- Added detailed sequence novelty analysis showing DiMA achieves 64.3% novelty on SwissProt while maintaining clustering diversity (CD₀.₉₅ = 0.992) (suggested by Reviewers bZ1a and sQ4r)
- Provided comprehensive breakdown of relationships between evaluation metrics (suggested by Reviewers bZ1a, sQ4r, and GJwR)
- Added biological relevance analysis showing 92% of generated sequences match known SUPERFAMILY annotations (suggested by Reviewers bZ1a and w8qc)

We also revised manuscript in accordance with the reviewers' suggestions. We have reorganized the structure with clear thematic blocks in the experiments section, clarified technical advantages of latent diffusion for protein generation, revised tables for consistent predentation of results, improved table and figure captions, changed citation style according to the guidelines.

These improvements address the main concerns raised by reviewers while maintaining the core technical contribution of our work: demonstrating that continuous latent diffusion can be effectively adapted for protein sequence generation through careful architectural choices and training procedures.

We are grateful to the reviewers for their time, effort and constructive suggestions. We believe the revisions have strengthened our paper and addressed the reviewers' concerns while maintaining its core technical contributions. We respectfully ask the reviewers to consider these improvements in their final assessment.

Best regards,

The Authors

---

### Meta-Review · Area_Chair_iU4d · 2024-12-20

**Metareview:**

After all discussion phases have concluded, three reviewers judge the paper to be marginally above the acceptance threshold, and three reviewers judge the paper to be marginally or clearly below the acceptance threshold.

Among other things, as positives the reviewers highlight useful ablation studies of the proposed method.
As areas of improvement reviewers highlighted the need for more thorough evaluation, an improved structure of the paper, correction of statements in the paper that were not deemed correct by the reviewers, removing too strong claims and a better discussion of related work. The authors have put in a substantial effort in their rebuttal to alleviate concerns, which has led to several increased scores, moving from [1, 6, 5, 3, 5, 3] to [5, 6, 6, 5, 6, 3]. Unfortunately, despite the productive discussion between reviewers and authors, the paper still needs some further improvement in terms of the evaluation of the method and the presentation of the paper for acceptance. I therefore recommend to reject this paper for this conference. I hope the authors can build on the feedback of the reviewers to further improve their manuscript and resubmit to another venue in future. Thank you to all reviewers and authors for engaging during the reviewing period.

**Additional Comments On Reviewer Discussion:**

See above.

---

### Decision · Program_Chairs · 2025-01-22

Reject